# Adaptive LoRA Experts Allocation and Selection for Federated Fine-Tuning

**Lei Wang** *
University of Florida
Gainesville, FL 32611
`leiwang1@ufl.edu`

**Jieming Bian** *
University of Florida
Gainesville, FL 32611
`jieming.bian@ufl.edu`

**Letian Zhang**
Middle Tennessee State University
Murfreesboro, TN 37132
`letian.zhang@mtsu.edu`

**Jie Xu**
University of Florida
Gainesville, FL 32611
`jie.xu@ufl.edu`

## Abstract

Large Language Models (LLMs) have demonstrated impressive capabilities across various tasks, but fine-tuning them for domain-specific applications often requires substantial domain-specific data that may be distributed across multiple organizations. Federated Learning (FL) offers a privacy-preserving solution, but faces challenges with computational constraints when applied to LLMs. Low-Rank Adaptation (LoRA) has emerged as a parameter-efficient fine-tuning approach, though a single LoRA module often struggles with heterogeneous data across diverse domains. This paper addresses two critical challenges in federated LoRA fine-tuning: 1. determining the optimal number and allocation of LoRA experts across heterogeneous clients, and 2. enabling clients to selectively utilize these experts based on their specific data characteristics. We propose FedLEASE (**Fed**erated adaptive **L**oRA **E**xpert **A**llocation and **SE**lection), a novel framework that adaptively clusters clients based on representation similarity to allocate and train domain-specific LoRA experts. It also introduces an adaptive top-$M$ Mixture-of-Experts mechanism that allows each client to select the optimal number of utilized experts. Our extensive experiments on diverse benchmark datasets demonstrate that FedLEASE significantly outperforms existing federated fine-tuning approaches in heterogeneous client settings while maintaining communication efficiency.

## 1 Introduction

Large Language Models (LLMs) have demonstrated remarkable capabilities across a wide range of tasks, from natural language understanding and generation to reasoning and problem solving [12, 38, 1, 39, 36]. Despite their impressive general abilities, these models often require fine-tuning to achieve optimal performance in domain-specific applications and specialized tasks [19]. Fine-tuning adapts pre-trained LLMs to particular domains, enhancing their performance on targeted tasks by incorporating domain-specific knowledge and patterns. This process has proven essential for applications in healthcare, finance, law, and science, where specialized expertise is required [9, 11]. However, effective fine-tuning typically requires large volumes of high-quality, domain-specific data, which may be distributed across multiple organizations. In many real-world scenarios, such data cannot be centralized due to privacy concerns or regulatory restrictions [43]. Federated Learning (FL) [31, 28, 27, 4, 41, 47, 22, 25] has emerged as a promising solution to this challenge, enabling

---

*The first two authors contributed equally to this work.

39th Conference on Neural Information Processing Systems (NeurIPS 2025).

collaborative model training across distributed data sources without sharing the raw data. In FL, clients train models locally using their private data and share only model updates with a central server, thereby preserving data privacy while leveraging the collective knowledge embedded in the distributed datasets.

The application of FL to LLM fine-tuning presents significant challenges due to the computational and communication constraints inherent in federated settings. Full fine-tuning of LLMs, which typically contain billions of parameters, is prohibitively expensive for many FL clients with limited resources. This has led to growing interest in Parameter-Efficient Fine-Tuning (PEFT) methods [19], which significantly reduce the number of trainable parameters. Among these PEFT approaches, Low-Rank Adaptation (LoRA) [20] has gained substantial traction due to its simplicity and effectiveness. LoRA introduces small trainable low-rank matrices alongside the frozen pre-trained weights, substantially reducing the number of parameters that need to be updated during fine-tuning in FL settings [23].

Although LoRA enables efficient domain adaptation, recent studies [15, 33] show that a single LoRA module often falls short in handling heterogeneous domains and complex tasks—especially in FL settings where clients hold data from distinct domains. Existing FL methods [45, 5, 35, 17] largely rely on a single shared LoRA module across all clients. While some personalized approaches [44] combine global and local LoRA modules to address heterogeneity, such binary designs cannot capture nuanced client similarities, where some clients share domain traits while others diverge significantly. This oversimplification leads to suboptimal knowledge sharing and underutilization of the collective learning potential. On the other hand, using too many LoRA experts—e.g., one per client—introduces computational overhead and risks representational collapse due to redundancy [8]. These competing constraints give rise to two key research questions: **(1)** ***Given heterogeneous client distributions, what is the optimal number of LoRA experts to allocate, and how should clients contribute to their training?*** Furthermore, client heterogeneity suggests that different clients may benefit from different expert combinations, leading to a second question: **(2)** ***Given allocated experts, how can each client dynamically determine the optimal number of experts to use based on its data characteristics?***

To address these questions, we conducted extensive empirical analysis with heterogeneous clients in a federated learning environment. Our analysis yielded two significant observations: **First**, clients with similar domain characteristics should collaboratively train shared LoRA experts, while clients with dissimilar data distributions should contribute to distinct experts. **Second**, different clients require different numbers of experts to achieve optimal performance, necessitating an adaptive approach to expert utilization rather than a fixed selection strategy. Based on these insights, we propose FedLEASE (short for **Fed**erated adaptive **L**ora **E**xpert **A**llocation and **SE**lection), a novel framework for federated LoRA fine-tuning that systematically addresses both research questions. For the ***first*** problem of optimal expert allocation, we introduce a principled data-driven approach that determines both how many experts are needed and which clients should collaborate on each expert. FedLEASE implements a brief initial training phase followed by mathematical clustering of clients based on their LoRA parameter similarity. This process leverages the silhouette coefficient to identify the optimal number of experts while ensuring clients with similar task characteristics contribute to shared experts. For the ***second*** problem, we introduce a novel adaptive top-$M$ mechanism that transforms the conventional Mixture-of-Experts (MoE) paradigm [21]. While traditional MoE approaches require manually specifying a fixed number of experts (top-k) for all inputs—a significant limitation in heterogeneous federated settings—our adaptive mechanism automatically determines the optimal number of experts for each client based on their specific data characteristics. Through an innovative router architecture that expands the output space from $\mathbb{R}^{M \times d}$ to $\mathbb{R}^{(2M-1) \times d}$, our approach enables dynamic expert selection ranging from a single expert to the full ensemble while guaranteeing the inclusion of each client's assigned expert. Together, these innovations create a comprehensive solution to the dual challenges of expert allocation and selection in federated LoRA fine-tuning. Our contributions can be summarized as follows:

- We identify and formalize two key challenges in federated LoRA fine-tuning: allocation of LoRA experts, and enabling clients to selectively utilize them based on data characteristics.

- We propose **FedLEASE**, a novel framework that clusters clients to train domain-specific LoRA experts and enables flexible expert selection via an adaptive top-$M$ MoE mechanism.

- Extensive experiments on diverse benchmarks demonstrate that FedLEASE consistently outperforms existing federated fine-tuning methods, achieving superior performance in heterogeneous settings while maintaining communication efficiency.

## 2 Related Works

**Parameter-Efficient Fine-Tuning.** Parameter-efficient fine-tuning (PEFT) reduces the cost of adapting large language models by updating only a small subset of parameters while freezing the rest [19]. Common PEFT techniques include adapters [13, 14], prefix-tuning [24, 26], and low-rank adaptation (LoRA) [20, 29]. LoRA injects trainable low-rank matrices into pre-trained weights, significantly cutting trainable parameters and computation. However, a single LoRA module can struggle with diverse domains and complex tasks [15, 29, 37], prompting Mixture-of-Experts (MoE) extensions that combine multiple small LoRA modules [29, 37]. Conversely, too many experts may introduce redundancy and collapse representations [8]. These centralized findings motivate our study of optimal LoRA deployment under heterogeneous data distributions in federated learning.

**PEFT in Federated Learning.** PEFT methods have become particularly suitable for resource-constrained federated learning settings by adjusting only a small number of lightweight parameters while keeping most pre-trained parameters unchanged. Various PEFT approaches have been integrated within FL frameworks [3], such as prompt-based fine-tuning [46, 18] and adapter-based tuning techniques [7, 6]. In this paper, we focus specifically on LoRA-based approaches in FL. FedIT [45] pioneered this direction by combining LoRA with the standard FedAvg algorithm, demonstrating its viability in distributed settings. Subsequent works like [42, 5] attempted to further enhance LoRA in FL by addressing challenges related to inexact server aggregation. Other research efforts [17] have investigated LoRA's application in data heterogeneous settings, but primarily focused on the relatively simpler label distribution non-IID scenario, where clients share the same underlying task but differ in their label distributions. Our work addresses a more complex and realistic scenario where clients may possess data from both similar and different tasks, representing true domain heterogeneity. Unlike prior works that typically employ a binary global-local architecture [44], we investigate the fundamental question of determining the optimal number and allocation of LoRA experts given heterogeneous client distributions. Additionally, while existing approaches apply the same aggregation strategy to all clients regardless of their data characteristics, our method adaptively determines client groupings and enables adaptive expert selection based on client-specific needs.

## 3 Preliminary and Motivation

### 3.1 LoRA and MoE Integration

Low-Rank Adaptation (LoRA) [20] has been proven to achieve comparable performance to full fine-tuning by inserting trainable low-rank matrices into each layer of a pre-trained model. For a pre-trained model with parameters $W_0 \in \mathbb{R}^{l \times d}$, where $d$ is the input dimension and $l$ is the output dimension, LoRA introduces two sequential low-rank matrices $A \in \mathbb{R}^{r \times d}$ and $B \in \mathbb{R}^{l \times r}$ to fit the residual weights for adaptation, where $r \ll \min(d, l)$. The forward computation is expressed as: $y = W_0 x + BAx$, where $A$ is typically initialized with random Gaussian values,

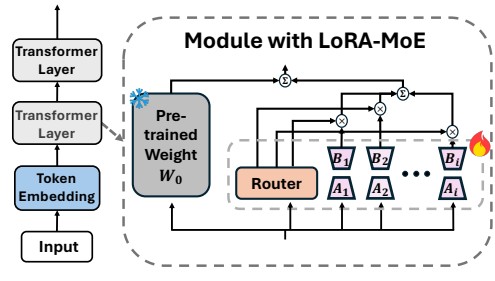

Figure 1: **Illustration of LoRA-MoE mechanism.**

while $B$ is initialized to zero to ensure a stable start to the fine-tuning process. Although LoRA performance is comparable to full fine-tuning in many scenarios, its effectiveness can significantly deteriorate when applied to heterogeneous data containing multiple tasks with different corpora. The performance gap between LoRA and full fine-tuning widens in such complex setting [2].

Recent research [37, 29] has explored integrating LoRA with MoE to address multi-domain adaptation challenges. In this integration, each expert in the MoE framework is implemented as a separate LoRA module rather than as a full neural network. A router network computes routing probabilities and the forward computation for such a LoRA-MoE system can be expressed as:

$$y = W_0 x + \sum_i p_i(x) \cdot B_i A_i x, \tag{1}$$

where $p_i(x)$ is the routing probability for expert $i$, and $A_i \in \mathbb{R}^{r \times d}$ and $B_i \in \mathbb{R}^{l \times r}$ are the low-rank matrices for the $i$-th LoRA expert. A extended top-$k$ MoE mechanism selects the LoRA experts

based on top-$k$ routing probabilities for each input, where $k$ is a fixed and pre-defined number. This integration offers significant advantages for handling diverse domains by leveraging different LoRA experts for different input types while maintaining the parameter efficiency of LoRA. In **centralized settings**, this approach has shown promising results for multi-domain adaptation [37]. However, applying LoRA-MoE in federated learning introduces unique challenges, particularly in determining the optimal number of experts, their allocation across heterogeneous clients, and how many experts each client should utilize based on their specific data.

### 3.2 Heterogeneous Federated Fine-tuning Scenario

Consider a system with $N$ clients, where each client $i \in \{1, 2, ..., N\}$ possesses a local dataset $\mathcal{D}_i = (x_j^i, y_j^i)_{j=1}^{|\mathcal{D}_i|}$, with each dataset potentially originating from similar or heterogeneous tasks. The goal of heterogeneous federated fine-tuning is to obtain models for each client that perform well on their respective data distributions. This can be formulated as the following optimization problem: $\min_{\mathcal{W}} \mathcal{L}(\mathcal{W}) = \sum_{i=1}^{N} \frac{|\mathcal{D}_i|}{|\mathcal{D}|} \mathcal{L}_i(W_i)$, where $\mathcal{L}_i$ is the local loss function for client $i$, $|\mathcal{D}| = \sum_{i=1}^{N} |\mathcal{D}_i|$ is the total size of data across all clients, and $\mathcal{W} = \{W_i\}_{i=1}^{N}$ denotes the set of fine-tuned models.

### 3.3 Observations

The objective of this work is to address two key problems of LoRA in complex heterogeneous federated learning settings: (1) determining the optimal number and allocation of LoRA experts, and (2) enabling each client to selectively utilize these experts according to their specific data characteristics. To investigate these issues, we conduct empirical studies that yield important insights.

**Observation 1: Clients with similar tasks/domains should contribute to the same LoRA expert through averaging, while those with different ones should be assigned to different LoRA experts.**

In realistic federated learning settings, client heterogeneity often goes beyond simple label distribution shifts and encompasses fundamental task differences. To examine this, we designed two experimental scenarios. **Scenario 1** involved four clients, each assigned a different GLUE task (SST-2, QNLI, MRPC, QQP) [40], representing a task-heterogeneous setting. **Scenario 2** used four clients all holding data from the same task (QNLI), thereby representing a task-homogeneous setting. For each scenario, we compared two methods: (1) **FedIT** [45], where each client trains a single shareable LoRA module that is averaged at the server, and (2) **FedLoRA-MoE**, where clients train individual LoRA modules without averaging. Instead, all modules are shared, and each client trains a MoE router to dynamically combine its own module with others'.

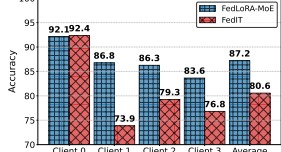

(a) **Scenario 1 task heterogeneity.** Each client holds data from different tasks (SST-2, QNLI, MRPC, QQP).

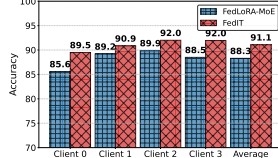

(b) **Scenario 2 task homogeneity.** Each client holds data from the same task (QNLI).

Figure 2: **Performance comparison between FedIT and FedLORA-MoE under two scenarios with different clients' task heterogeneity.**

As shown in Figure 2, our experimental results indicate that in Scenario 1, **FedIT** performs significantly worse than **FedLoRA-MoE**, suggesting that clients with different tasks struggle to contribute effectively to a single shared LoRA module. This observation is consistent with findings in centralized settings [15, 37], where a single LoRA module proves insufficient for handling diverse domains. Conversely, in Scenario 2, FedIT outperforms FedLoRA-MoE, suggesting that using separate LoRA experts for homogeneous clients may be redundant and can degrade performance while increasing inference overhead, which aligns with results from [8]. These findings offer key insights into expert allocation strategies under varying task heterogeneity in FL.

**Observation 2: Task heterogeneity among clients can be detected through representation similarity of LoRA $B$ matrices after brief local training.**

We observe that task similarity between clients can be effectively assessed by computing the cosine similarity of their LoRA $B$ matrices after a short period of local training. To validate this, we conducted an experiment with four clients: two using the SST-2 dataset and two using the QNLI

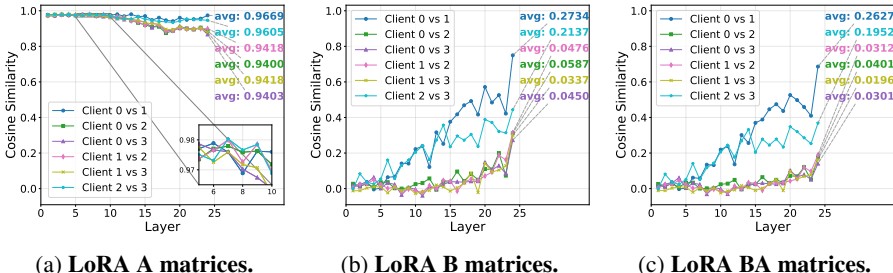

(a) **LoRA A matrices.**    (b) **LoRA B matrices.**    (c) **LoRA BA matrices.**

Figure 3: **Visualization result of cosine similarity among clients using different LoRA matrices.** Note that client 0 and 1 hold data from **SST-2** and client 2 and 3 hold data from **QNLI**.

dataset. As shown in Figure 3, clients working on the same task develop highly similar LoRA $B$ matrices, while those working on different tasks exhibit significantly lower similarity. Interestingly, this pattern is exclusive to the LoRA $B$ matrices; the $A$ matrices show no consistent relationship with task similarity. This observation supports findings in [37, 17], which suggest that the output transformation matrix $B$ captures task-specific information, whereas the input matrix $A$ tends to encode general linguistic features shared across tasks. Although a similar task-specific pattern can be observed by analyzing the product $BA$, this requires matrix multiplication and full-rank projection recovery, which incurs significantly higher computational cost. In contrast, using the $B$ matrices alone provides a lightweight yet effective proxy for task similarity, making it more practical in FL.

**Observation 3: Clients utilize varying numbers of LoRA experts for optimal performance.**

To investigate our second research question, we conducted additional experiments under the task-heterogeneous setting (Scenario 1) using the **FedLoRA-MoE** approach. We varied the top-$k$ parameter in the MoE router, testing values of $k = 2, 3, 4$ (where $k = 4$ corresponds to using all available experts). As shown in Figure 4, different clients achieve optimal performance at different $k$ values—some benefit most from $k = 2$, while others perform better with $k = 3$ or $k = 4$. This result highlights that even if the total number of trained LoRA experts is fixed, clients have varying needs regarding how many experts they should utilize. A static top-$k$ selection strategy is therefore suboptimal across all clients. These findings motivate the design of an *adaptive expert selection mechanism* that dynamically determines the optimal number of experts for each client based on its specific data characteristics.

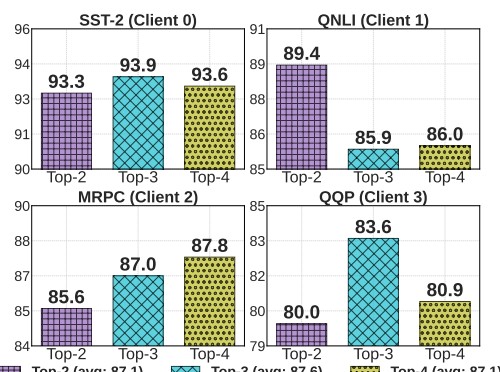

Figure 4: **Comparison on accuracy of clients using different top-$k$ strategies under the task-heterogeneous setting.**

## 4   Proposed Method

In this section, we describe the framework design of our proposed method FedLEASE by explaining how it addresses the two important challenges in complex federated fine-tuning settings. A **theoretical analysis** of the proposed method can be found in Section F.

### 4.1   Adaptive LoRA Experts Allocation

A fundamental challenge in federated LoRA fine-tuning is determining the optimal number of experts and identifying which clients should contribute to each expert. We address this challenge through a systematic data-driven approach that analyzes similarity patterns in client-specific adaptations. Our method begins with a brief initialization phase where each client $i \in \{1, 2, \ldots, N\}$ independently trains a LoRA module $(A_i, B_i)$ for $E$ epochs using its local dataset $\mathcal{D}_i$. This phase serves to capture initial task-specific adaptations in the LoRA parameters. Upon completion, each client transmits its LoRA parameters to the central server.

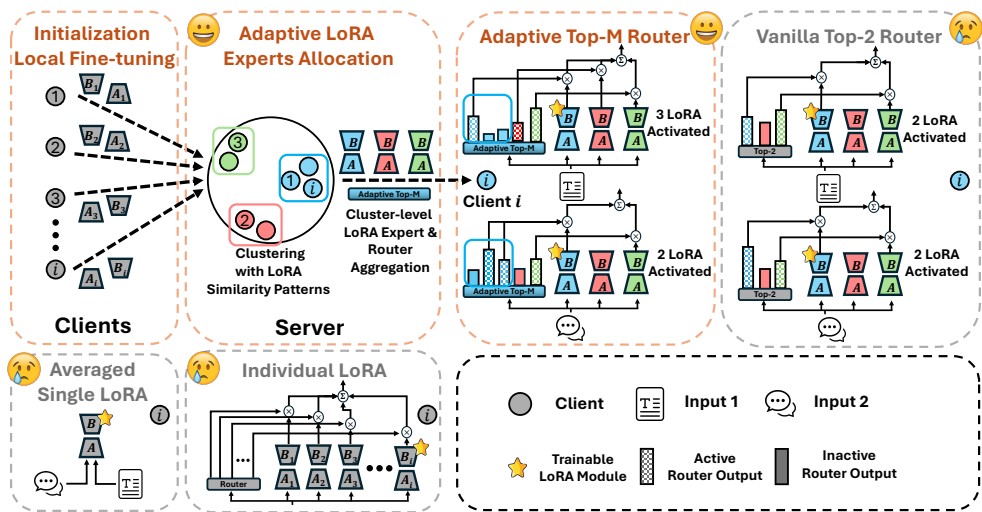

Figure 5: **Illustration of our proposed adaptive LoRA experts allocation and top-$M$ experts selection mechanism. Average Singe LoRA** and **Individual LoRA** shows the LoRA experts allocation strategies employed by FedIT and FedLoRA-MoE respectively as described in Section 3.3. **Vanilla Top-2 Router** is an example of the MoE-based fixed top-$k$ LoRA experts selection strategy.

Based on the observations in Section 3.3, the $B_i$ matrices exhibit distinct similarity patterns that align with underlying task relationships. Accordingly, we define a distance metric between clients $i$ and $j$ using cosine similarity across all layers:

$$d(i,j) = \frac{1}{|L|} \sum_{l \in L} \left( 1 - \frac{\mathbf{B}_i^l \cdot \mathbf{B}_j^l}{\|\mathbf{B}_i^l\| \cdot \|\mathbf{B}_j^l\|} \right), \tag{2}$$

where $L$ is the set of model layers, $\mathbf{B}_i^l$ is the flattened $B_i$ matrix at layer $l$, and $\|\cdot\|$ denotes the Euclidean norm. To identify the optimal expert allocation, we systematically evaluate all potential clustering configurations. For each possible number of clusters $k \in \{2, 3, \ldots, M_{max}\}$, where $M_{max}$ should be set to satisfy the maximum number of LoRA modules given the limited computation resources of clients, we apply Agglomerative Hierarchical Clustering [32] to partition clients $\mathcal{C}^k = Cluster(\{B_i\}_{i=1}^N, d, k)$, resulting in $k$ clusters $\mathcal{C}^k = \{C_1^k, C_2^k, \ldots, C_k^k\}$.

We evaluate the quality of each clustering configuration using the silhouette coefficient [34], which measures how well each client fits within its assigned cluster relative to others. The average silhouette score for a $k$-cluster configuration is defined as $S(k) = \frac{1}{N} \sum_{i=1}^N s^k(i)$, where $s^k(i)$ is the silhouette coefficient for client $i$ given by $s^k(i) = \frac{b^k(i) - a^k(i)}{\max(a^k(i), b^k(i))}$. Here, $a^k(i)$ denotes the average dissimilarity between client $i$ and all other clients in the same cluster (intra-cluster cohesion), while $b^k(i)$ is the minimum average dissimilarity between client $i$ and clients in other clusters (inter-cluster separation).

The optimal number of experts $M = \underset{2 \leq k \leq M_{max}}{\arg\max} S(k)$ is selected as the $k$ that maximizes the average score. This approach ensures that we identify both the optimal number of experts needed and the most coherent grouping of clients based on their adaptation patterns. A high silhouette score indicates that clients within the same cluster exhibit similar adaptation characteristics, while clients in different clusters show distinct patterns. Once the optimal clustering $\mathcal{C}^M = \{C_1^M, C_2^M, \ldots, C_M^M\}$ is determined, we initialize each LoRA by aggregating the clients within the corresponding cluster:

$$A_j^{\text{expert}} = \frac{1}{|C_j^M|} \sum_{i \in C_j^M} A_i, \quad B_j^{\text{expert}} = \frac{1}{|C_j^M|} \sum_{i \in C_j^M} B_i \tag{3}$$

The server then distributes the experts along with their cluster assignment information. This approach effectively balances between having too few experts (which would fail to capture domain diversity) and too many experts (which would lead to redundancy and inefficient parameter usage).

## 4.2 Adaptive top-M LoRA Experts

Having addressed the first critical question of expert allocation in the previous section, we now turn our attention to the second challenge: how can each client selectively utilize these experts based on its specific data characteristics? This question is particularly important as our observation 3 demonstrated that different clients require different numbers of experts to achieve optimal performance.

After receiving the allocated LoRA experts, each client enters the main training phase with access to all $M$ experts. During this phase, each client only updates its assigned LoRA expert while leveraging knowledge from other experts to enhance performance on its local data distribution. The challenge lies in determining how many and which experts each client should selectively utilize, as a fixed top-$k$ selection strategy proves suboptimal across heterogeneous clients. A standard Mixture-of-Experts approach with top-$k$ selection would formulate the forward computation as:

$$y = W_0 x + \sum_{i \in \text{TopK}(\omega, k)} \omega_i B_i A_i x, \tag{4}$$

where $\omega = (\omega_1, \ldots, \omega_M)$ denotes the routing weights computed as $\omega_i = \text{softmax}(G_i x)$ with $G_i \in \mathbb{R}^{M \times d}$ being the trainable router.

This standard approach, however, presents two limitations in our context: (1) it requires manual tuning of $k$ for each client, which is impractical in federated settings, and (2) it does not guarantee the inclusion of the client's assigned expert, which is essential for effective parameter updates. We address these limitations through an innovative adaptive routing mechanism that ensures the client's assigned expert is always selected while dynamically determining the appropriate number of additional experts to utilize. Instead of employing a conventional router with dimensions $\mathbb{R}^{M \times d}$, we expand the router's output space to $\mathbb{R}^{(2M-1) \times d}$, where the **first $M$ outputs** are connected to the client's assigned expert, while the **remaining $M - 1$ outputs** correspond to the other experts.

Formally, our adaptive routing mechanism is expressed as:

$$y = W_0 x + \sum_{i \in \text{TopK}(\hat{\omega}, M)} \hat{\omega}_i \cdot \begin{cases} B_j A_j x, & \text{if } i < M \\ B_{i-M+1} A_{i-M+1} x, & \text{if } i \geq M \end{cases}, \tag{5}$$

where $\hat{\omega} = \text{softmax}(G_i x) \in \mathbb{R}^{2M-1}$ and $j$ denotes the expert index assigned to the client.

An illustration of the proposed **adaptive top-$M$ mechanism** is shown in Figure 5. The proposed router $\hat{\omega} \in \mathbb{R}^{2M-1}$ allows each client to decide, for every input, how many and which experts contribute, instead of relying on a globally fixed $k$. When the top-ranked scores lie among the first $M$ entries of $\hat{\omega}$, the computation is dominated by the client's own assigned expert $E_j$. When large scores appear in the remaining $M-1$ positions, the client leverages additional experts. Intermediate cases arise naturally: if the router selects $p$ of the first $M$ entries (i.e., $p$ internal components of the assigned expert $E_j$) together with $M-p$ entries from the other experts, then in effect $M-p+1$ *unique experts* participate in the forward computation.

To make the mechanism more concrete, consider a case with $M=3$ experts $\{E_1, E_2, E_3\}$ and a client whose assigned expert is $E_1$. The router produces

$$\hat{\omega} = [\; \underbrace{\hat{\omega}_1^{E_1}, \hat{\omega}_2^{E_1}, \hat{\omega}_3^{E_1}}_{\text{connected to assigned expert } E_1}, \quad \underbrace{\hat{\omega}_4^{E_2}, \hat{\omega}_5^{E_3}}_{\text{connected to other experts} E_2, E_3} \;]$$

where the first three entries are independently learned routing scores corresponding to distinct internal components of the assigned expert $E_1$ (all routed to $B_1 A_1 x$), and the last two correspond to the other experts $E_2$ and $E_3$ (routed to $B_2 A_2 x$ and $B_3 A_3 x$). With different input samples inducing different routing score distributions, our proposed top-$M$ mechanism can accordingly select different expert combinations:

- **Sample 1:** if the top-3 are $\hat{\omega}_1^{E_1}, \hat{\omega}_2^{E_1}, \hat{\omega}_3^{E_1}$, then only the assigned expert $E_1$ contributes.
- **Sample 2:** if the top-3 are $\hat{\omega}_1^{E_1}, \hat{\omega}_2^{E_1}, \hat{\omega}_4^{E_2}$, then two unique experts $\{E_1, E_2\}$ are selected.
- **Sample 3:** if the top-3 are $\hat{\omega}_2^{E_1}, \hat{\omega}_4^{E_2}, \hat{\omega}_5^{E_3}$, then all three experts $\{E_1, E_2, E_3\}$ participate.

This mechanism guarantees that each client's designated expert $E_j$ always participates—facilitating stable local updates—while allowing flexible, data-driven cooperation with other experts. By enabling

the router to balance the internal components of $E_j$ with the contributions from the remaining $\{E_m\}_{m \neq j}$, the method adapts to both input complexity and inter-client heterogeneity, avoiding any manual tuning of $k$ and achieving effective expert utilization for federated fine-tuning.

### 4.3 Algorithm Workflow

FedLEASE operates in two phases: initialization and iterative training. A detailed Algorithm 1 can be found in Section A.

**Server Operations.** In the initialization phase, the server clusters clients based on the similarity of their initial $B_i$ matrices using the silhouette-based method in Section 4.1, yielding $M$ clusters $\mathcal{C} = \{C_1, \ldots, C_M\}$. For each cluster $C_j$, expert parameters $(A_j^{\text{expert}}, B_j^{\text{expert}})$ are initialized by averaging LoRA modules within the cluster and distributed to all clients along with cluster assignments.

In each communication round, the server receives updated expert parameters and router networks from clients and performs within-cluster aggregation. The updated experts and routers are then broadcast to clients for the next training round.

**Client Operations.** Each client $i$ begins with $E$ epochs of local fine-tuning to obtain $(A_i, B_i)$. After clustering, the client receives the full set of experts and its assigned cluster ID. During local training, client $i$ updates only its assigned expert and the corresponding router $G_j^i \in \mathbb{R}^{(2M-1) \times d}$, keeping all other experts fixed. Using the adaptive top-$M$ strategy in Section 4.2, each client dynamically determines how many experts to utilize, ranging from just one to all $M$, based on its local data. Upon completion, only the updated expert and router are uploaded to the server.

## 5 Experiment

In this section, we evaluate the performance of FedLEASE, against baseline methods on two types of datasets: natural language understanding (NLU) and natural language generation (NLG).

### 5.1 Training Details

For the NLU task, we use RoBERTa [30] as the pre-trained model and fine-tune it on the GLUE benchmark [40]. For the NLG task, we adopt LLaMA2 [39] as the pre-trained model and fine-tune it on the FLAN dataset [10].

**NLU Task.** We consider 16 clients in total, with four clients assigned to each of the four GLUE datasets. Each client's data is randomly partitioned from the corresponding full dataset. RoBERTa-Large (355M) [30] (24 transformer layers) from HuggingFace is used as the base model. AdamW is adopted as the optimizer for all methods, with a batch size of 128, local epochs set to 2, and a total of 25 communication rounds. Following [35], LoRA is applied to the query and value projections in the attention layers, and the classification head is frozen after initialization. For our method, the upper bound of experts $M_{max}$ is set to 8 and the LoRA rank to 4. Baselines are configured to ensure comparable computational workloads. Learning rates are selected via grid search from $\eta \in \{1E-4, 3E-4, 5E-4, 1E-3, 3E-3, 5E-3\}$. Accuracy is utilized as the evaluation metric.

**NLG Tasks.** For NLG tasks, we use LLaMA-2-7B [39] with 8-bit quantization (32 transformer layers) from Hugging Face as the base model and select four FLAN datasets—Text Editing, Struct to Text, Sentiment Analysis, and Commonsense Reasoning—to construct a heterogeneous client setting. A total of 8 clients are considered, with each dataset assigned to two clients. Each client has 600 training samples and 200 test samples. All methods use AdamW as the optimizer, with a batch size of 8, local epochs set to 2, 10 communication rounds and the upper bound of experts $M_{max}$ is set to 8. LoRA is applied to the query and value matrices in the attention layers, with a LoRA rank of 8. Learning rates are selected via grid search from $\eta \in \{1E-4, 3E-4, 1E-3, 3E-3, 1E-2\}$. Followed by [44], we choose ROUGE-1 as the evaluation metric.

**Baseline Methods.** To assess the effectiveness of FedLEASE, we compare it with the following state-of-the-art federated LoRA fine-tuning methods: **FedIT** [45], **FedSA** [17], **FFA-LoRA** [35], and **FedDPA** [44]. Additionally, we include a clustered federated learning method, IFCA [16], and adapt it with LoRA fine-tuning as a baseline, denoted as **IFCA + LoRA**.

## 5.2 Natural Language Understanding

We evaluate our method on four GLUE [40] benchmark datasets: SST-2, QNLI, MRPC, and QQP. Unlike prior works such as FedSA, which assume clients differ only in label distribution, we adopt a more realistic heterogeneous setting, where each client is assigned a different dataset from the four. The setup includes 16 clients in total, with four clients per dataset. Each client's data is randomly partitioned from the full corresponding dataset. We use RoBERTa-Large (355M) [30] from the HuggingFace library as the base model. Additional training details are provided in Section 5.1.

**Performance Comparison.** Table 1 summarizes the results on the NLU tasks. FedLEASE consistently outperforms all baselines, both in terms of average performance and on each individual dataset. Notably, it achieves an average improvement of 3.16% over the strongest baseline across the four GLUE tasks. These gains are attributed to two key innovations: (1) clients with similar data distributions are grouped to collaboratively train a shared expert, while clients with distinct data contribute to different LoRA experts; and (2) an adaptive top-$M$ expert selection strategy that enables each client to personalize expert usage based on its local data. Although IFCA+LoRA incorporates client clustering, it falls short of FedLEASE due to its lack of cross-cluster knowledge transfer, resulting in isolated learning and reduced generalization. In contrast, FedLEASE allows clients to dynamically leverage experts, facilitating effective cross-task knowledge sharing. These improvements are achieved without additional computational or communication overhead, highlighting the scalability and efficiency of FedLEASE.

**Ablation on Adaptive Expert Allocation.** FedLEASE assigns 16 clients to 4 experts based on clustering results (illustrated in Section B). To evaluate the impact of expert training allocation, we compare against the following alternatives: *FedLoRA-Single*: A single expert is trained with contributions from all clients. To ensure fair comparison, we test two variants with

Table 2: **Ablation on adaptive expert allocation.**

| Method | # Experts | % Param | Performance (%) |
|---|---|---|---|
| FedLoRA-Single ($r = 4$) | 1 | 0.1106% | 82.00 |
| FedLoRA-Single ($r = 16$) | 1 | 0.4426% | 83.84 |
| FedLoRA-Individual | 16 | 0.3320% | 80.69 |
| FedLEASE (w/o adaptive top-$M$) | 4 | 0.1383% | 85.91 |
| **FedLEASE (Ours)** | 4 | 0.2075% | **87.76** |

different LoRA ranks: FedLoRA-Single ($r = 4$) and FedLoRA-Single ($r = 16$). *FedLoRA-Individual*: Each client trains a separate expert, resulting in 16 experts and one-to-one client-expert mapping. *FedLEASE (w/o adaptive top-$M$)*: Uses the same expert allocation as FedLEASE but replaces adaptive top-$M$ router with vanilla fixed top-2. Results in Table 2 show that our clustering-based expert allocation achieves the best performance, even without the adaptive top-$M$ mechanism. Both FedLoRA-Single variants underperform due to limited capacity to model heterogeneous data, while FedLoRA-Individual, despite higher computational cost, still lags behind FedLEASE, validating the effectiveness of our clustering strategy in balancing knowledge sharing and task specificity.

We also evaluate **router aggregation strategies**. Compared to maintaining individual routers per client, our approach—averaging router networks within each group—achieves better performance (Figure 6a), confirming the benefit of shared routing among clients with similar data.

**Ablation on Adaptive top-$M$ Mechanism.** We assess the effectiveness of our adaptive top-$M$ mechanism by comparing it with fixed top-$k$ strategies (top-1 through top-4). As shown in Figure 6b, different clients achieve optimal performance with different $k$ values—for example, clients with QQP dataset perform best with top-2, whereas those with MRPC dataset benefit more from top-4. Notably, all fixed top-$k$ strategies except top-1 achieve comparable performance and outperform baseline methods reported in Table 1, also highlighting the importance of the expert allocation strategy. In contrast, our adaptive top-$M$ mechanism consistently surpasses all fixed strategies across clients, demonstrating its capability to dynamically select the optimal number of experts per input. To further illustrate this adaptivity, Figure 6c visualizes expert selection across layers for 16 clients in our main experiment. The patterns show substantial variation both across clients and across layers within the

Table 1: **Performance on GLUE dataset (RoBERTa-Large-355M).**

| Methods | % Param | SST-2 | QNLI | MRPC | QQP | Average | $\Delta$ |
|---|---|---|---|---|---|---|---|
| FedIT [45] | 0.2213% | $93.33 \pm 0.38$ | $85.43 \pm 1.41$ | $76.35 \pm 2.58$ | $73.82 \pm 4.01$ | $82.23 \pm 2.10$ | - |
| FFA-LoRA [35] | 0.1107% | $90.32 \pm 0.83$ | $77.53 \pm 2.18$ | $78.45 \pm 0.84$ | $77.95 \pm 2.15$ | $81.06 \pm 1.50$ | -1.17 |
| FedDPA [44] | 0.2213% | $91.90 \pm 0.43$ | $83.13 \pm 0.69$ | $81.60 \pm 1.61$ | $81.35 \pm 1.22$ | $84.49 \pm 0.99$ | +2.26 |
| FedSA [17] | 0.2213% | $91.97 \pm 0.81$ | $82.70 \pm 0.53$ | $82.08 \pm 1.51$ | $81.65 \pm 1.37$ | $84.60 \pm 1.05$ | +2.37 |
| IFCA+LoRA [16] | 0.2213% | $92.95 \pm 0.50$ | $85.90 \pm 0.64$ | $78.63 \pm 2.38$ | $80.42 \pm 1.30$ | $84.48 \pm 1.21$ | +2.25 |
| **FedLEASE** | 0.2075% | $\mathbf{93.33 \pm 0.30}$ | $\mathbf{87.22 \pm 1.16}$ | $\mathbf{86.93 \pm 0.68}$ | $\mathbf{83.57 \pm 0.96}$ | $\mathbf{87.76 \pm 0.78}$ | **+5.53** |

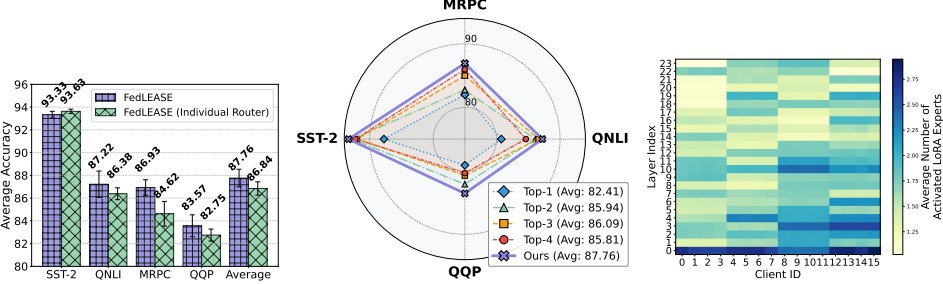

(a) **Comparison on router aggregation strategies.** (b) **Comparison between adaptive top-$M$ and top-$k$ mechanisms.** (c) **Visualization results of expert selection.**

Figure 6: **Ablation on router aggregation strategies and adaptive top-$M$ mechanism.**

same client—some layers rely on a single expert, while others combine multiple. Client ID 0-3, 4-7, 8-11 and 12-15 correspond to SST-2, QNLI, MRPC and QQP, respectively. Specifically, we can observe that: **1. Vertical trend:** As layers deepen, the average number of activated LoRA experts tends to increase; **2. Horizontal trend:** As task difficulty increases (reflected by decreasing accuracy from SST-2 to QQP in Table 1), the average number of activated LoRA experts also increases; **3. Cluster-level:** Since we aggregate the router within each cluster group during server aggregation, rather than maintaining each client's individual one, clients within the same cluster tend to exhibit similar but not identical expert selection patterns. This fine-grained adaptivity underscores the limitations of fixed top-$k$ approaches and confirms the necessity of our adaptive top-$M$ mechanism.

We also conduct sensitivity analysis to evaluate the robustness of our method under various settings. Due to space constraints, results examining the effects of local epochs, LoRA rank, number of clients, data heterogeneity, and the expert upper bound $M_{\max}$ are provided in Section C.

## 5.3   Natural Language Generation

In addition to NLU tasks, we evaluate our method on NLG tasks. We adopt LLaMA-2-7B [39] as the base model and use four FLAN datasets—Text Editing, Struct to Text, Sentiment Analysis, and Commonsense Reasoning—to construct a heterogeneous client setting. We consider 8 clients in total, with each dataset assigned to

Table 3: **Performance on FLAN dataset (LLaMA-2-7B).**

| Methods | % Param | Text Editing | Struct to Text | Sentiment Analysis | Commonsense Reasoning | Average |
|---|---|---|---|---|---|---|
| FedIT [45] | 0.0622% | 59.30 | 52.14 | 43.95 | 73.95 | 57.33 |
| FFA-LoRA [35] | 0.0311% | 59.37 | 50.86 | 41.23 | 72.61 | 56.02 |
| FedDPA [44] | 0.0622% | 65.30 | 53.40 | 47.68 | 72.84 | 59.81 |
| FedSA [17] | 0.0622% | 64.82 | 54.48 | 46.70 | 74.81 | 60.20 |
| IFCA + LoRA [16] | 0.0622% | 66.56 | 53.46 | 46.17 | 72.73 | 59.73 |
| **FedLEASE (Ours)** | 0.0584% | **67.08** | **54.94** | **48.13** | **76.66** | **61.70** |

two clients. Training details are provided in Section 5.1. As shown in Table 3, FedLEASE consistently outperforms all baselines on NLG tasks. Compared to the strongest overall baseline, FedLEASE achieves gains of 2.26%, 0.46%, 1.43%, and 1.85% on Text Editing, Struct to Text, Sentiment Analysis, and Commonsense Reasoning, respectively. On average, FedLEASE improves by 1.50%, highlighting its ability to handle heterogeneous client data across both classification and generation tasks, demonstrating its generalizability beyond NLU.

## 6   Conclusion

We presented FedLEASE, a novel framework addressing key challenges in federated LoRA fine-tuning for heterogeneous clients. Our approach combines intelligent client clustering for optimal expert allocation with an adaptive top-$M$ mechanism that dynamically determines expert selection based on client-specific needs. Extensive experiments on NLU and NLG tasks demonstrate that FedLEASE consistently outperforms existing approaches across diverse datasets while maintaining communication efficiency. Our method effectively balances knowledge sharing and domain specificity. Future work could explore dynamic clustering techniques, additional parameter-efficient fine-tuning methods, and further communication optimizations for resource-constrained federated settings.

## Acknowledgments

The work of Lei Wang, Jieming Bian and Jie Xu is partially supported by NSF under grants 2433886, 2505381 and 2515982. The work of Letian Zhang is partially supported by NSF under grant 2348279 and also supported by MTSU Stark Land project.

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

# A  The Algorithm of FedLEASE

The algorithm of proposed FedLEASE is summarized in Algorithm 1.

---
**Algorithm 1** FedLEASE: Federated Low-Rank Expert Learning

---
1: **Initialization Phase:**
2: Server initializes model parameters $\{\mathbf{A}_j, \mathbf{B}_j\}_{j=1}^M$
3: **for** each client $i \in \{1, \ldots, N\}$ **do**
4:     Client $i$ performs local training on $(\mathbf{A}_i, \mathbf{B}_i)$ for $E$ epochs
5:     Client $i$ sends trained parameters $(\mathbf{A}_i, \mathbf{B}_i)$ to the server
6: **end for**
7: Server computes distance $d(i, j)$ using cosine similarity:
8:     $d(i, j) = \frac{1}{|L|} \sum_{l \in L} \left( 1 - \frac{\mathbf{B}_i^l \cdot \mathbf{B}_j^l}{\|\mathbf{B}_i^l\| \|\mathbf{B}_j^l\|} \right)$
9: Server determines optimal number of experts $M$ using silhouette scores
10: Server clusters clients using Agglomerative Hierarchical Clustering:
11:     $\{C_1, \ldots, C_M\} \leftarrow \text{Cluster}(\{\mathbf{B}_i\}_{i=1}^N, d, M)$
12: Server aggregates expert parameters per cluster:
13:     $\mathbf{A}_j^{\text{expert}} \leftarrow \frac{1}{|C_j|} \sum_{i \in C_j} \mathbf{A}_i, \mathbf{B}_j^{\text{expert}} \leftarrow \frac{1}{|C_j|} \sum_{i \in C_j} \mathbf{B}_i$
14: **Iterative Training Phase:**
15: **for** each communication round $t = 1, 2, \ldots, T$ **do**
16:     **for** each client $i \in \{1, \ldots, N\}$ with $i \in C_j$ **do**
17:         Client $i$ receives all expert parameters $\{(\mathbf{A}_k^{\text{expert}}, \mathbf{B}_k^{\text{expert}})\}_{k=1}^M$
18:         Client $i$ uses local router $\mathbf{G}_i \in \mathbb{R}^{(2M-1) \times d}$
19:         Client $i$ trains assigned expert $j$ parameters and router $\mathbf{G}_i$ locally
20:         Compute adaptive routing with weights $\hat{\omega} \leftarrow \text{softmax}(\mathbf{G}_i x) \in \mathbb{R}^{2M-1}$
21:         Output:
22:         $y \leftarrow \mathbf{W}_0 x + \sum_{p \in \text{TopK}(\hat{\omega}, M)} \hat{\omega}_p \cdot \begin{cases} \mathbf{B}_j^{\text{expert}} \mathbf{A}_j^{\text{expert}} x, & \text{if } p < M \\ \mathbf{B}_{p-M+1}^{\text{expert}} \mathbf{A}_{p-M+1}^{\text{expert}} x, & \text{if } p \geq M \end{cases}$
23:         Client $i$ uploads updated parameters $(\mathbf{A}_j^i, \mathbf{B}_j^i)$ to server
24:     **end for**
25:     **for** each expert $j = 1, \ldots, M$ **do**
26:         Server aggregates expert parameters:
27:         $\mathbf{A}_j^{\text{expert}} \leftarrow \frac{1}{|C_j|} \sum_{i \in C_j} \mathbf{A}_j^i, \mathbf{B}_j^{\text{expert}} \leftarrow \frac{1}{|C_j|} \sum_{i \in C_j} \mathbf{B}_j^i$
28:     **end for**
29: **end for**

---

# B  Clustering Results

Our clustering analysis reveals natural groupings of clients based on the cosine similarity of their LoRA B matrices. Figure 7 presents a comprehensive visualization of the clustering results using three complementary approaches.

Figure 7(a) shows the Silhouette scores for different numbers of clusters, ranging from $k = 2$ to $k = 8$. The Silhouette score, which measures how similar objects are to their assigned cluster compared to other clusters, peaks at $k = 4$, indicating that 4 is the optimal number of clusters for the main experiments setting. This finding suggests that clients naturally form 4 distinct groups based on their model adaptations. In Figure 7(b), we visualize the distance matrix derived from cosine similarity between client LoRA B matrices. The heatmap reveals clear block diagonal structures, which further supports the existence of distinct client clusters. The darker squares along the diagonal represent groups of clients with high intra-cluster similarity, while lighter colors indicate greater dissimilarity between different clusters.

Finally, Figure 7(c) presents a hierarchical clustering dendrogram based on the same cosine similarity measure. The dendrogram provides an alternative view of client relationships, illustrating how clients

progressively merge into larger groups. The vertical axis represents the distance at which clusters are combined, with longer vertical lines indicating greater separation between clusters.

These clustering results provide strong evidence for natural groupings of clients, suggesting an optimal clustering of clients into 4 distinct groups. To ensure robustness of our findings, all results presented are averaged across 5 independent runs with different random initializations.

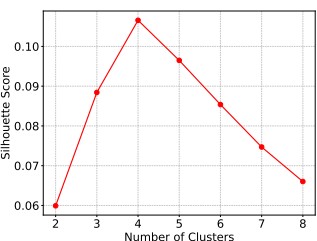 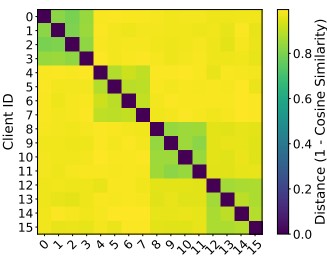 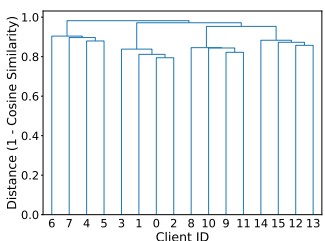

(a) Silhouette scores for different numbers of clusters, showing optimal clustering at 4 clusters.

(b) Heatmap of distance matrix derived from cosine similarity between client LoRA B matrices.

(c) Hierarchical clustering dendrogram based on cosine similarity of client LoRA B matrices.

Figure 7: **Visualization of client clustering results based on cosine similarity of LoRA B matrices.**

While clustering is indeed an essential component of our proposed method—playing a key role in expert allocation based on client similarity—the specific choice of clustering algorithm is not the focus of our contribution. Our observations in Section 3.3 suggest that as long as the method captures pairwise similarity between clients (e.g., via LoRA B matrices), the overall performance is relatively robust to the particular clustering strategy.

We adopt Agglomerative Hierarchical Clustering due to its ability to operate directly on pairwise distances without requiring pre-defined centroids. To validate the generality of our approach, we also applied Spectral Clustering, which similarly supports pairwise similarity inputs, and observed comparable performance. Tables 4 and 5 below demonstrate that both clustering methods achieve similar silhouette scores and downstream performance, reinforcing that our performance gains stem primarily from the expert allocation and adaptive top- selection mechanisms, rather than the specific clustering algorithm used.

Table 4: **Silhouette Scores Comparison**

| Clustering Method | 2 | 3 | 4 | 5 | 6 | 7 | 8 |
|---|---|---|---|---|---|---|---|
| Spectral Clustering | 0.0585 | 0.0820 | **0.1023** | 0.0739 | 0.0637 | 0.0549 | 0.0218 |
| Agglomerative Hierarchical Clustering | 0.0599 | 0.0884 | **0.1066** | 0.0965 | 0.0854 | 0.0747 | 0.0660 |

Table 5: **Performance Comparison**

| Clustering Method | SST2 | QNLI | MRPC | QQP | Average |
|---|---|---|---|---|---|
| Spectral Clustering | 93.97 | 86.63 | 86.48 | 83.40 | 87.62 |
| Agglomerative Hierarchical Clustering | 93.33 | 87.22 | 86.93 | 83.57 | 87.76 |

## C  Sensitivity Analysis

In this section, we perform multiple sensitivity analysis to demonstrate the robustness of our proposed method under different settings.

We vary the number of local training epochs to examine its effect on performance. The results in Figure 8 show that our proposed method consistently outperforms all baseline methods across different local epoch settings ($E = 5$), confirming that FedLEASE's effectiveness is not dependent on specific epoch configurations.

**Impact of LoRA Rank.** We test the performance with different LoRA ranks, adjusting the rank parameter for both our method and baselines. As shown in Table 6, FedLEASE maintains superior

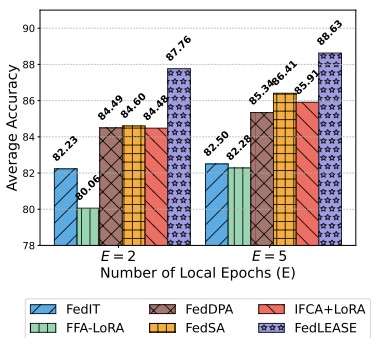

Figure 8: **Impact of Local Epochs.**

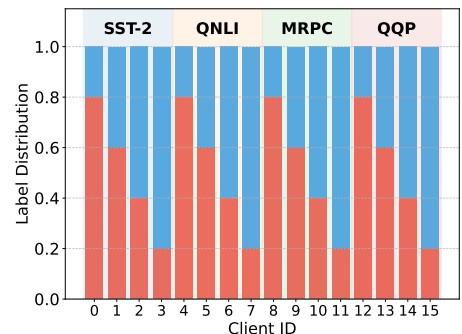

Figure 9: **Illustration of Non-IID Label Distribution.**

performance across all tested rank values. The performance gap is particularly notable at lower ranks, highlighting our method's efficiency in parameter utilization.

Table 6: **Impact of Rank.**

| Methods | $r = 2$ | | | | | $r = 6$ | | | | |
|---|---|---|---|---|---|---|---|---|---|---|
| | SST-2 | QNLI | MRPC | QQP | Average | SST-2 | QNLI | MRPC | QQP | Average |
| FedIT [45] | 93.20 | 80.37 | 77.35 | 78.95 | 82.47 | 92.97 | 83.88 | 80.68 | 77.45 | 83.74 |
| FFA-LoRA [35] | 91.83 | 74.48 | 74.10 | 77.10 | 79.38 | 91.63 | 75.50 | 78.07 | 74.60 | 79.95 |
| FedDPA [44] | 92.63 | 81.80 | 80.93 | 80.30 | 83.91 | 91.65 | 82.25 | 83.18 | 78.52 | 83.90 |
| FedSA [17] | 92.05 | 82.13 | 78.68 | 80.42 | 83.32 | 92.05 | 82.72 | 81.10 | 80.75 | 84.16 |
| IFCA+LoRA [16] | 93.33 | 84.95 | 77.05 | 81.82 | 84.29 | 93.40 | 85.90 | 79.60 | 82.38 | 85.32 |
| **FedLEASE** | **93.80** | **87.32** | **84.63** | **83.30** | **87.26** | **93.43** | **86.12** | **87.42** | **84.10** | **87.77** |

Table 7: **Impact of Number of Clients.**

| Methods | $N = 8$ | | | | | $N = 32$ | | | | |
|---|---|---|---|---|---|---|---|---|---|---|
| | SST-2 | QNLI | MRPC | QQP | Average | SST-2 | QNLI | MRPC | QQP | Average |
| FedIT [45] | 93.50 | 79.63 | 80.70 | 78.25 | 83.02 | 92.72 | 81.65 | 76.67 | 72.70 | 80.94 |
| FFA-LoRA [35] | 91.70 | 71.10 | 82.23 | 75.15 | 80.05 | 90.42 | 79.82 | 73.53 | 77.12 | 80.22 |
| FedDPA [44] | 92.13 | 84.17 | 84.22 | 80.60 | 85.28 | 91.65 | 78.30 | 73.75 | 79.37 | 80.77 |
| FedSA [17] | 92.65 | 84.35 | 77.10 | 78.93 | 83.26 | 92.25 | 81.12 | 75.45 | 78.73 | 81.89 |
| IFCA+LoRA [16] | 93.82 | 85.83 | 81.07 | 79.92 | 85.16 | **93.75** | 84.25 | 81.15 | 78.73 | 84.47 |
| **FedLEASE** | **93.92** | **86.88** | **87.14** | **82.95** | **87.72** | 93.25 | **87.50** | **86.07** | **81.70** | **87.13** |

**Impact of Client Numbers.** To evaluate scalability, we change the number of clients in the system to 8 and 32. Table 7 demonstrates that our proposed method maintains its performance advantage with different number of clients, indicating robust scalability to federated networks.

**Impact of Data Heterogeneity.** We evaluate our method across varying degrees of data heterogeneity, focusing on realistic task heterogeneity rather than simple label distribution shifts. We consider three unbalanced task distribution settings: (1) Least heterogeneous: 16 clients having 2 kinds of NLU datasets (10 with QNLI and 6 with QQP); (2) Mildly heterogeneous: 16 clients having 3 kinds of NLU datasets (4 with SST-2, 7 with QNLI, 5 with QQP); and (3) Most heterogeneous: 16 clients having 4 kinds of NLU datasets (3 with SST2, 6 with QNLI, 2 with MRPC, 5 with QQP). The results in Table 8 demonstrate that FedLEASE consistently outperforms baseline methods across all heterogeneity levels, with the performance advantage becoming more pronounced as heterogeneity increases.

We further conduct the additional experiments under both task and label non-i.i.d. setting, and the label distribution is illustrated in Figure 9. Note the task distribution is same as what we used in the main experiment. As shown in Table 9, our proposed mthod still outperforms other baselines in this both task and label Non-IID setting.

Table 8: **Impact of Task Heterogeneity.**

| Methods | Least Heterogeneous | | | Mildly Heterogeneous | | | | Most Heterogeneous | | | | |
|---|---|---|---|---|---|---|---|---|---|---|---|---|
| | QNLI (10) | QQP (6) | Avg. | SST-2 (4) | QNLI (7) | QQP (5) | Avg. | SST-2 (3) | QNLI (6) | MRPC (2) | QQP (5) | Avg. |
| FedIT [45] | 87.72 | 71.78 | 79.75 | 92.75 | 88.51 | 69.76 | 83.67 | 92.10 | 85.22 | 71.80 | 77.74 | 81.71 |
| FFA-LoRA [35] | 86.02 | 77.58 | 81.80 | 91.65 | 86.19 | 69.98 | 82.61 | 76.20 | 81.53 | 70.30 | 75.18 | 75.80 |
| FedDPA [44] | 84.20 | 81.85 | 83.03 | 91.78 | 83.67 | 79.00 | 84.82 | **93.50** | 82.37 | 79.90 | 83.32 | 84.77 |
| FedSA [17] | 83.74 | 80.35 | 82.05 | 91.10 | 80.07 | 80.50 | 83.89 | 91.50 | 81.92 | **83.10** | 77.92 | 83.61 |
| IFCA+LoRA [16] | 87.75 | 78.22 | 82.98 | 93.50 | 87.77 | 82.16 | 87.81 | 92.50 | 86.75 | 77.50 | 84.16 | 85.23 |
| **FedLEASE** | **87.77** | **84.18** | **85.98** | **93.85** | **89.23** | **84.66** | **89.25** | 92.93 | **88.60** | 83.05 | **84.74** | **87.33** |

Table 9: **Performance under Task and Label Non-IID.**

| Methods | SST-2 | QNLI | MRPC | QQP | Average |
|---|---|---|---|---|---|
| FedIT [45] | 93.37 | 83.12 | 80.73 | 72.25 | 82.37 |
| FFA-LoRA [35] | 89.77 | 74.93 | 76.70 | 79.72 | 80.28 |
| FedDPA [44] | 86.28 | 78.48 | 77.15 | 77.18 | 79.77 |
| FedSA [17] | 87.25 | 77.85 | 73.50 | 74.43 | 78.26 |
| IFCA+LoRA [16] | 92.95 | 83.88 | 80.07 | 79.00 | 83.98 |
| **FedLEASE (Ours)** | **93.60** | **84.13** | **81.92** | **80.05** | **84.93** |

Table 10: **Performance Comparison of Different Model Configurations.**

| Expert Upper Bound | Final Number of Experts | SST-2 | QNLI | MRPC | QQP | Average |
|---|---|---|---|---|---|---|
| $M_{max} = 2$ | 2 | 92.15 | 85.70 | 82.08 | 82.03 | 85.49 |
| $M_{max} = 3$ | 3 | 93.73 | 87.05 | 84.55 | 83.50 | 87.21 |
| $M_{max} = 4$ | 4 | 93.62 | 87.65 | 87.05 | 83.07 | 87.85 |
| $M_{max} = 8$ (Used) | 4 | 93.33 | 87.22 | 86.93 | 83.57 | 87.76 |

Then we perform additional experiments under label non-i.i.d. setting and follow the exact setup used in FedSA under a Dirichlet ($\alpha = 0.5$) partitionaing scheme using 20 clients on the QQP dataset. Results in Tables 11 to 14 demonstrate that FedLEASE consistently outperforms existing methods even under label-heterogeneous conditions, further confirming the robustness and generality of our proposed approach.

Table 11: **Label Distribution under Dirichlet(0.5) on QQP**

| Client ID | 0 | 1 | 2 | 3 | 4 | 5 | 6 | 7 | 8 | 9 | 10 | 11 | 12 | 13 | 14 | 15 | 16 | 17 | 18 | 19 |
|---|---|---|---|---|---|---|---|---|---|---|---|---|---|---|---|---|---|---|---|---|
| Class 0 (%) | 0.7 | 18.1 | 70.3 | 8.0 | 0.4 | 85.8 | 92.3 | 18.7 | 90.9 | 33.2 | 99.9 | 20.6 | 81.8 | 85.9 | 24.6 | 9.8 | 80.5 | 92.1 | 93.1 | 75.2 |
| Class 1 (%) | 99.3 | 81.9 | 29.7 | 92.0 | 99.6 | 14.2 | 7.7 | 81.3 | 9.1 | 66.8 | 0.1 | 79.4 | 18.2 | 14.1 | 75.4 | 90.2 | 19.5 | 7.9 | 6.9 | 24.8 |

Table 12: **Silhouette Scores under Only Label Non-IID**

| Clusters | 2 | 3 | 4 | 5 | 6 | 7 | 8 |
|---|---|---|---|---|---|---|---|
| Score | **0.1829** | 0.1139 | 0.0553 | 0.0532 | 0.0482 | 0.0527 | 0.0506 |

Table 13: **Cluster Groups under Only Label Non-IID**

| Group | Clients |
|---|---|
| Group 0 | 2, 5, 6, 8, 10, 12, 13, 16, 17, 18, 19 |
| Group 1 | 0, 1, 3, 4, 7, 9, 11, 14, 15 |

**Impact of Expert Upper Bound** $M_{max}$**.** In our main experiments, we set the expert upper bound to $M_{max} = 8$. To investigate the sensitivity of our approach to this parameter, we conducted additional experiments with lower upper bounds ($M_{max} = 2, 3, 4$) using the same experimental setup: 16 clients with data from 4 GLUE datasets. As shown in Table 10, we observe comparable performance between $M_{max} = 8$ and $M_{max} = 4$ configurations. This aligns with our clustering analysis in Section B, which indicates that this particular configuration requires only 4 experts. This finding validates that our method can efficiently determine the appropriate number of experts needed even when the budget ($M_{max}$) exceeds the system's actual requirements. When restricted to $M_{max} < 4$, we observe a performance degradation compared to the $M_{max} = 4$ or $M_{max} = 8$ settings, further confirming the importance of allocating an adequate number of experts. Nevertheless, it is noteworthy that even with this constrained expert budget, our proposed method still outperforms all baseline methods. This demonstrates the robustness of our approach and its ability to make efficient use of even limited expert resources through effective allocation and adaptive selection.

# D    Computational Overhead

Our clustering step is performed only once during the initialization phase and is not repeated during iterative training. Thus, its runtime impact is negligible. As observed in Section 3.3, using only the LoRA B matrices offers an efficient and lightweight proxy for task similarity, given their small size compared to full model weights or BA products. As shown in Table 15, we measured the clustering time (3.11 seconds on Intel Xeon Platinum 8570 CPU), which is significantly shorter than the total training time (193.49 seconds with local training on the NVIDIA B200 GPU):

Table 14: **Performance under Only Label Non-IID**

| Method | FedIT | FFA-LoRA | FedDPA | FedSA | IFCA+LoRA | FedLEASE |
|---|---|---|---|---|---|---|
| Accuracy (%) | 83.52 | 83.05 | 85.78 | 86.85 | 84.73 | **89.23** |

Table 15: Comparison of Computing Time

| Time (s) | FedIT | FFA-LoRA | FedDPA | FedSA | IFCA+LoRA | FedLEASE |
|---|---|---|---|---|---|---|
| Local Per-Epoch Training Time | 3.75 | 3.41 | 3.82 | 3.72 | 3.85 | 3.78 |
| Global Aggregation Time | 0.048 | 0.038 | 0.051 | 0.042 | 0.061 | 0.055 |
| Clustering Time | - | - | - | - | 2.77 | 3.11 |
| Total Training Time | 188.70 | 171.45 | 192.28 | 187.05 | 263.28 | 193.49 |

# E  Limitations

While FedLEASE achieves strong performance in heterogeneous federated fine-tuning, it has limitation. The current framework assumes a static client population and fixed expert assignments throughout training. In practical federated environments where client availability and data distributions evolve over time, this rigidity may limit adaptability. Future work could explore dynamic clustering or meta-routing strategies to accommodate such non-stationary conditions.

# F  Convergence Analysis

In this section, we analyze the convergence properties of our proposed FedLEASE method.

We define the following notation:

- $\mathcal{C} = \{\mathcal{C}_1, \mathcal{C}_2, ..., \mathcal{C}_M\}$ represents the partition of clients into $M$ clusters
- $\theta_i^t = \{A_i^t, B_i^t, G_i^t\}$ denotes the trainable parameters for client $i$ at round $t$
- $\Theta_j^t = \{A_j^{expert,t}, B_j^{expert,t}, G_j^{expert,t}\}$ denotes the aggregated parameters for cluster $j$ at round $t$
- For any client $i \in \mathcal{C}_j$, we define $j(i) = j$ as the cluster it belongs to

The cluster-level parameters are computed by averaging the parameters of all clients in the cluster:

$$\Theta_j^t = \frac{1}{|\mathcal{C}_j|} \sum_{i \in \mathcal{C}_j} \theta_i^t \tag{6}$$

We denote the client-level loss function as $f_i(\theta_i | \{\Theta_k\}_{k=1}^M)$ and the cluster-level loss function as:

$$F_j(\Theta_j | \{\Theta_k\}_{k \neq j}) = \frac{1}{|\mathcal{C}_j|} \sum_{i \in \mathcal{C}_j} f_i(\theta_i | \{\Theta_k\}_{k=1}^M) \tag{7}$$

## F.1  Assumptions

To establish convergence, we make the following assumptions:

**Assumption F.1** (Client-Level Smoothness)**.** For each client $i$, the loss function $f_i$ is $\mu$-smooth with respect to $\theta_i$, i.e., for any $\theta_i^1, \theta_i^2$:

$$\|\nabla f_i(\theta_i^1) - \nabla f_i(\theta_i^2)\| \leq \mu \|\theta_i^1 - \theta_i^2\| \tag{8}$$

**Assumption F.2** (Client-Level Strong Convexity)**.** For each client $i$, the loss function $f_i$ is $\lambda$-strongly convex with respect to $\theta_i$, i.e., for any $\theta_i^1, \theta_i^2$:

$$f_i(\theta_i^2) \geq f_i(\theta_i^1) + \langle \nabla f_i(\theta_i^1), \theta_i^2 - \theta_i^1 \rangle + \frac{\lambda}{2} \|\theta_i^2 - \theta_i^1\|^2 \tag{9}$$

**Assumption F.3** (Bounded Expert Sensitivity). The optimal parameters for client $i$ are sensitive to changes in the expert parameters with a bounded Lipschitz constant. For any two sets of expert parameters $\{\Theta_k^1\}_{k=1}^M$ and $\{\Theta_k^2\}_{k=1}^M$:

$$\|\theta_i^*(\{\Theta_k^1\}) - \theta_i^*(\{\Theta_k^2\})\| \leq \beta \sum_{k=1}^M \|\Theta_k^1 - \Theta_k^2\| \tag{10}$$

where $\theta_i^*(\{\Theta_k\})$ represents the optimal parameters for client $i$ given fixed expert parameters.

**Assumption F.4** (Cluster Assignment Stability). After the initial clustering phase, the assignment of clients to clusters remains stable throughout the training process.

**Theorem F.5** (Convergence of FedLEASE). *With the assumptions of client-level smoothness, client-level strong convexity, limited inter-cluster influence, and cluster assignment stability, we can derive:*

$$\|\Theta_j^{t+1} - \Theta_j^t\| \leq \frac{2\epsilon}{1 - \beta M} + (\beta M)^t \max_j \|\Theta_j^1 - \Theta_j^0\|$$

*If $\beta M < 1$ and the local training at each round converges to a neighborhood of the optimal solution, then the sequence of cluster models $\{\Theta_j^t\}$ generated by FedLEASE converges to a stable point for each cluster $j$.*

## F.2 Proof

At round $t$, each client $i$ performs local training to update its parameters. Let $\theta_i^{t,s}$ represent the parameters of client $i$ after $s$ steps of local training within round $t$.

The gradient descent update rule for client $i$ at step $s$ is:

$$\theta_i^{t,s+1} = \theta_i^{t,s} - \eta \nabla f_i(\theta_i^{t,s}|\{\Theta_k^t\}_{k=1}^M) \tag{11}$$

From Assumption 1 (client-level smoothness), we have:

$$f_i(\theta_i^{t,s+1}) \leq f_i(\theta_i^{t,s}) + \langle \nabla f_i(\theta_i^{t,s}), \theta_i^{t,s+1} - \theta_i^{t,s} \rangle + \frac{\mu}{2} \|\theta_i^{t,s+1} - \theta_i^{t,s}\|^2 \tag{12}$$

$$= f_i(\theta_i^{t,s}) - \eta \|\nabla f_i(\theta_i^{t,s})\|^2 + \frac{\mu\eta^2}{2} \|\nabla f_i(\theta_i^{t,s})\|^2 \tag{13}$$

$$= f_i(\theta_i^{t,s}) - \eta(1 - \frac{\mu\eta}{2}) \|\nabla f_i(\theta_i^{t,s})\|^2 \tag{14}$$

Let $\theta_i^* = \theta_i^*(\{\Theta_k^t\})$ denote the optimal parameters for client $i$ given the fixed expert parameters at round $t$. From Assumption 2 (client-level strong convexity), we have:

$$f_i(\theta_i^*) \geq f_i(\theta_i^{t,s}) + \langle \nabla f_i(\theta_i^{t,s}), \theta_i^* - \theta_i^{t,s} \rangle + \frac{\lambda}{2} \|\theta_i^* - \theta_i^{t,s}\|^2 \tag{15}$$

Then we can establish:

$$\|\nabla f_i(\theta_i^{t,s})\|^2 \geq 2\lambda(f_i(\theta_i^{t,s}) - f_i(\theta_i^*)) \tag{16}$$

Substituting this into our earlier inequality:

$$f_i(\theta_i^{t,s+1}) - f_i(\theta_i^*) \leq f_i(\theta_i^{t,s}) - f_i(\theta_i^*) - \eta(1 - \frac{\mu\eta}{2}) \|\nabla f_i(\theta_i^{t,s})\|^2 \tag{17}$$

$$\leq f_i(\theta_i^{t,s}) - f_i(\theta_i^*) - 2\eta\lambda(1 - \frac{\mu\eta}{2})(f_i(\theta_i^{t,s}) - f_i(\theta_i^*)) \tag{18}$$

$$= (1 - 2\eta\lambda(1 - \frac{\mu\eta}{2}))(f_i(\theta_i^{t,s}) - f_i(\theta_i^*)) \tag{19}$$

Denoting $\rho = (1 - 2\eta\lambda(1 - \frac{\mu\eta}{2}))$, with properly chosen learning rate $\eta < \frac{2}{\mu}$, we have $\rho \in (0, 1)$. By recursively applying this inequality for $s$ steps:

$$f_i(\theta_i^{t,s}) - f_i(\theta_i^*) \leq \rho^s(f_i(\theta_i^{t,0}) - f_i(\theta_i^*)) \tag{20}$$

From strong convexity (Assumption 2), we can relate the optimality gap in function value to the parameter distance:

$$\frac{\lambda}{2}\|\theta_i^{t,s} - \theta_i^*\|^2 \leq f_i(\theta_i^{t,s}) - f_i(\theta_i^*) \tag{21}$$

Therefore:

$$\|\theta_i^{t,s} - \theta_i^*\|^2 \leq \frac{2\rho^s}{\lambda}(f_i(\theta_i^{t,0}) - f_i(\theta_i^*)) \tag{22}$$

With a sufficient number of local training steps $S$, we can ensure:

$$\|\theta_i^{t+1} - \theta_i^*(\{\Theta_k^t\})\| \leq \epsilon_i \tag{23}$$

where $\theta_i^{t+1} = \theta_i^{t,S}$ and $\epsilon_i$ can be made arbitrarily small by increasing $S$.

This establishes that each client's model converges to an approximate optimal solution for fixed expert parameters.

Now we analyze the stability of cluster models across communication rounds. For a cluster $j$, with the assumption of Cluster Assignment Stability, the aggregated parameters after round $t$ are:

$$\Theta_j^{t+1} = \frac{1}{|\mathcal{C}_j|}\sum_{i\in\mathcal{C}_j}\theta_i^{t+1} \tag{24}$$

The difference between consecutive cluster models is:

$$\|\Theta_j^{t+1} - \Theta_j^t\| = \left\|\frac{1}{|\mathcal{C}_j|}\sum_{i\in\mathcal{C}_j}\theta_i^{t+1} - \frac{1}{|\mathcal{C}_j|}\sum_{i\in\mathcal{C}_j}\theta_i^t\right\| \tag{25}$$

$$\leq \frac{1}{|\mathcal{C}_j|}\sum_{i\in\mathcal{C}_j}\|\theta_i^{t+1} - \theta_i^t\| \tag{26}$$

For each client $i \in \mathcal{C}_j$, we have:

$$\|\theta_i^{t+1} - \theta_i^t\| \leq \|\theta_i^{t+1} - \theta_i^*(\{\Theta_k^t\})\| + \|\theta_i^*(\{\Theta_k^t\}) - \theta_i^*(\{\Theta_k^{t-1}\})\| + \|\theta_i^*(\{\Theta_k^{t-1}\}) - \theta_i^t\| \tag{27}$$

$$\leq \epsilon_i + \beta\sum_{k=1}^{M}\|\Theta_k^t - \Theta_k^{t-1}\| + \epsilon_i \tag{28}$$

$$= 2\epsilon_i + \beta\sum_{k=1}^{M}\|\Theta_k^t - \Theta_k^{t-1}\| \tag{29}$$

Let $\epsilon = \max_i \epsilon_i$ and $\Delta^t = \max_j \|\Theta_j^{t+1} - \Theta_j^t\|$. Substituting into our cluster difference bound:

$$\|\Theta_j^{t+1} - \Theta_j^t\| \leq \frac{1}{|\mathcal{C}_j|}\sum_{i\in\mathcal{C}_j}(2\epsilon + \beta\sum_{k=1}^{M}\|\Theta_k^t - \Theta_k^{t-1}\|) \tag{30}$$

$$= 2\epsilon + \beta\sum_{k=1}^{M}\|\Theta_k^t - \Theta_k^{t-1}\| \tag{31}$$

$$\leq 2\epsilon + \beta M\Delta^{t-1} \tag{32}$$

Taking the maximum over all clusters:

$$\Delta^t \leq 2\epsilon + \beta M\Delta^{t-1} \tag{33}$$

When $\beta M < 1$, this is a contraction, and by iterating:

$$\Delta^t \leq 2\epsilon\sum_{i=0}^{t-1}(\beta M)^i + (\beta M)^t\Delta^0 \tag{34}$$

$$\leq \frac{2\epsilon}{1 - \beta M} + (\beta M)^t\Delta^0 \tag{35}$$

As $t \to \infty$, $\Delta^t \to \frac{2\epsilon}{1-\beta M}$, which can be made arbitrarily small by increasing local training steps (reducing $\epsilon$).

Our analysis demonstrates that FedLEASE converges at both client and cluster levels:

1. Each client's model converges to an approximate optimal solution with error bounded by $\epsilon_i$
2. The cluster models stabilize with a maximum change between rounds bounded by $\frac{2\epsilon}{1-\beta M}$

The convergence is guaranteed when:

- The learning rate is appropriately chosen ($\eta < \frac{2}{\mu}$)
- The inter-cluster influence is limited ($\beta M < 1$)
- Sufficient local training steps are performed (to reduce $\epsilon$)

