# OpenReview forum: "Adaptive LoRA Experts Allocation and Selection for Federated Fine-Tuning"
_NeurIPS.cc/2025/Conference — NeurIPS 2025 poster_

### Official Review · Reviewer_VvHY · 2025-06-21

**Clarity:** 3
**Significance:** 2
**Originality:** 3
**Rating:** 4
**Confidence:** 4

**Summary:**

The authors propose a novel PEFT methods for federated learning, FedLEASE. They consider two key challenge: the optimal number and allocation of LoRA experts in heterogeneous clients, how clients choose to use these LoRA. FedLEASE firstly clusters clients based on LoRA similarity, and then adopts an adaptive top-M MoE mechanism to determine the local LoRA number. Experiments are conducted to evaluate the performance of the proposed mechanism.

**Questions:**

Please refers to weakness.

**Ethical Concerns:**

["NO or VERY MINOR ethics concerns only"]

**Final Justification:**

Thanks for the authors' detailed response. My concerns have been addressed.

**Limitations:**

yes

**Quality:**

3

**Strengths And Weaknesses:**

Strength:
1. The structure of this paper is clear and reasonable.

2. This paper is easy to follow and generally well presented.

3. The proposed method shows impressive accuracy improvement on the datasets.


Weakness:
1. Clustering based on LoRA parameter similarity makes less of a difference than traditional FL clustering based on the whole model’s parameters. The special feature of the B matrix has been discovered in the FedSA.

2. The motivation and setting of heterogeneity are strange. Why do clients of different tasks need to collaborate? General FL considers label, feature, and model heterogeneity. However, contrary to the method's motivation, the experiments adopt different datasets in a benchmark with the same task.

3. Implement details of the experiments should be in the main paper.

4. The citation is not standard enough, such as FedDPA [37] is in the proceedings of NeurIPS 2024. The published version should be cited.

---

> ### Author Rebuttal · Authors · 2025-07-31
>
> We appreciate your insightful comments!
>
> **W1: Use of LoRA B for Clustering vs. Prior FL Clustering Approaches**
>
> While FedSA also recognizes the distinct roles of LoRA $A$ and $B$ matrices, its method primarily uses this insight for *selective aggregation*—aggregating only the $A$ matrices while keeping $B$ matrices local to preserve client-specific information.
>
> In contrast, **our approach uniquely leverages the LoRA $B$ matrices for clustering clients into distinct groups**, a fundamentally different purpose from FedSA. We apply hierarchical clustering on the $B$ matrices to capture task similarity and dynamically allocate personalized LoRA experts, which enables expert-aware routing and fine-tuning in federated settings.
>
> Beyond differentiating from FedSA, our method also offers clear advantages over traditional FL clustering methods like IFCA. Traditional clustering FL methods clusters clients using full model updates or losses, which incurs significant computational and communication cost. In contrast, FedLEASE uses only the small LoRA $B$ matrices, which are significantly more lightweight yet sufficient for capturing task-level differences. Moreover, our method couples this clustering with an **adaptive top-$M$ expert selection mechanism**, allowing sample-level flexibility across experts—something IFCA lacks.
>
> As shown in our empirical results (§3.3), clustering based solely on LoRA $B$ matrices yields comparable result with LoRA $BA$-based clustering, with far less overhead. Our experiments further show that FedLEASE outperforms IFCA+LoRA by a clear margin in both performance and efficiency.
>
> Therefore, while prior works have considered either clustering (e.g., IFCA) or LoRA-specific aggregation (e.g., FedSA), we think **FedLEASE is the first to both address expert allocation and utilization challenges by LoRA $B$-based clustering with flexible top-$M$ expert selection**, providing a novel and effective solution for federated LoRA fine-tuning..
>
> ---
>
> **W2: Justification for Task-Heterogeneous Setting**
>
> While we acknowledge that traditional federated learning often adopts Dirichlet partitioning to simulate label non-IID settings, our task-heterogeneous setup is explicitly designed to better reflect real-world scenarios of federated fine-tuning for large language models.
>
> In practical deployments, clients typically possess data from fundamentally different domains and tasks, rather than working on the same task with skewed labels. For example:
> - Customer service platforms with dialogue transcripts
> - Educational institutions with question-answering datasets
> - Medical research labs with clinical trial summaries
> - Legal departments with contract summarization tasks
> - E-commerce companies with product review classification
>
> Such **task-level heterogeneity** introduces more **challenging but realistic federated conditions** compared to simple label imbalance, and has also been adopted in prior work such as FedDPA (NeurIPS 2024).
>
> In our experiments, we emulate this realistic setup by using benchmarks such as:
> - **GLUE**, which combines diverse NLU tasks including sentiment analysis (SST-2), natural language inference (QNLI), paraphrase detection (QQP), and textual similarity (MRPC);
> - and the **Flan Collection**, which aggregates NLG tasks across question answering, summarization, reasoning, and more.
>
> These benchmarks are not a single dataset with label skew, but rather **collections of distinct tasks**, validating our claim of **task heterogeneity** in both motivation and implementation.
>
> Moreover, we also conducted evaluations of our method under **label non-IID conditions**, with results presented in **Figure 9** and **Table 7**. The findings demonstrate that FedLEASE maintains superior performance compared to all baseline methods, even when operating under these data distribution imbalances.
>
> ---
>
> **W3&W4: Implement details and citation format**
>
> We thank the reviewer for these helpful suggestions. We agree that placing critical implementation details in the main paper will enhance readability and reproducibility, and we will reorganize accordingly in the final version. Additionally, we acknowledge that certain citations, such as FedDPA, should reference their official published versions. We will correct and standardize all citations in the final manuscript to ensure accuracy and clarity.

---

> > ### Comment · Reviewer_VvHY · 2025-08-05
> >
> > Thanks for the authors' detailed response. My concerns have been addressed.

---

### Official Review · Reviewer_j5zx · 2025-06-27

**Clarity:** 3
**Significance:** 3
**Originality:** 3
**Rating:** 5
**Confidence:** 3

**Summary:**

This paper introduces FedLEASE, a novel framework for federated fine-tuning of large language models (LLMs) using adaptive LoRA. FedLEASE tackles two key challenges in federated settings: (1) how to allocate LoRA experts across heterogeneous clients, and (2) how clients can selectively use relevant experts. The method clusters clients by representation similarity and assigns domain-specific LoRA experts accordingly. It also employs an adaptive top-M Mixture-of-Experts mechanism to optimize expert selection per client. Experiments show that FedLEASE outperforms existing methods on diverse datasets, achieving better performance under data heterogeneity while maintaining communication efficiency.

**Questions:**

1) System Overheads (Computation, Memory, Communication): The proposed FedLEASE framework introduces additional mechanisms such as expert clustering, multiple LoRA experts, and dynamic expert selection. However, the paper does not provide a quantitative analysis of the resulting overheads.

* Can the authors elaborate on the computational, memory, and communication overheads introduced by FedLEASE?

* Specifically, how do these overheads compare to other baselines?

* Providing metrics such as peak memory usage, training time, and convergence rate (e.g., in wall-clock time or communication rounds) would help assess practical deployability.

2) Adaptivity and Sensitivity of the Cluster Count (M): The framework relies on clustering experts, with the number of clusters M which is fixed.

* Can the authors discuss the implications of using a static value for M?

* Is there a principled way to choose or adapt M dynamically based on the client distribution or similarity structure?

* If fully adaptive selection is not feasible, an empirical sensitivity analysis showing how performance varies with different M values would be highly valuable.

**Ethical Concerns:**

["NO or VERY MINOR ethics concerns only"]

**Final Justification:**

Thank you for your thoughtful responses, which answered my questions well.

**Limitations:**

yes

**Quality:**

3

**Strengths And Weaknesses:**

Strengths
+ Valuable empirical insights: The paper offers several meaningful insights drawn from rigorous experimentation. It shows that training a single shared LoRA module is ineffective under heterogeneous data distributions, while training completely separate modules is suboptimal for homogeneous settings. The study also highlights the effectiveness of using only the LoRA B matrix for representation similarity and demonstrates that allowing a variable number of experts per client leads to improved performance.

+ Comprehensive evaluation on accuracy: The experiments are thorough, covering a range of datasets and tasks with relevant baseline comparisons. The inclusion of ablation studies in the appendix adds depth and supports the claims. The overall evaluation convincingly demonstrates the effectiveness of the proposed FedLEASE framework in diverse federated settings.

Weaknesses
- Lack of quantitative analysis/evaluation on efficiency metrics: While accuracy is well covered, the evaluation lacks quantitative results on key efficiency metrics such as computational overhead, memory usage, and time-to-accuracy. For example, reporting convergence in terms of both communication rounds and wall-clock time would help assess the practical cost and scalability of FedLEASE, especially in real-world deployments with resource-constrained clients.

---

> ### Author Rebuttal · Authors · 2025-07-31
>
> Thank you for your meaningful suggestions!
>
> **W1: System Overheads**
> Regarding the training and clustering overhead, the clustering step is performed **only once** during the initialization phase and is not repeated during iterative training. Thus, its runtime impact is negligible. As described in §3.3, using only the **LoRA B matrices** provides an efficient and lightweight proxy for task similarity, due to their significantly smaller size compared to full model weights or LoRA A–B combinations.
>
> To further support this, we measured the clustering time as **3.11** seconds on an Intel Xeon Platinum 8570 CPU, which is substantially shorter than the total training time (**193.49** seconds with local training on an NVIDIA B200 GPU). The detailed timing comparisons are provided below:
>
> | Time (s)                        | FedIT | FFA-LoRA | FedDPA | FedSA  | IFCA+LoRA | FedLEASE |
> |---------------------------------|-------|----------|--------|--------|------------|----------|
> | Local Per-Epoch Training Time   | 3.75  | 3.41     | 3.82   | 3.72   | 3.85       | 3.78     |
> | Global Aggregation Time         | 0.048 | 0.038    | 0.051  | 0.042  | 0.061      | 0.055    |
> | Clustering Time | –     | –        | –      | –      | 2.77         | 3.11     |
> | Total Training Time             | 188.70| 171.45   | 192.28 | 187.05 | 263.28     | 193.49   |
>
> We also add a table summarizing convergence speed, demonstrating that FedLEASE achieves superior accuracy under the same round.
>
> **Accuracy Performance Across Training Rounds (%)**
>
> | **Method**     | **Round 1** | **Round 6** | **Round 11** | **Round 16** | **Round 21** | **Round 25** |
> |----------------|-------------|-------------|--------------|--------------|--------------|--------------|
> | FedIT          | 51.02       | 57.68       | 71.57        | 77.45        | 82.23        | 82.23        |
> | FFA-LoRA       | 50.66       | 59.84       | 70.53        | 77.79        | 81.08        | 81.06        |
> | FedDPA         | 51.33       | 59.36       | 75.99        | 81.94        | 84.23        | 84.49        |
> | FedSA          | 51.24       | 59.41       | 76.25        | 81.39        | 84.71        | 84.60        |
> | IFCA+LoRA      | 51.31       | 54.20       | 74.00        | 82.12        | 84.29        | 84.48        |
> | **FedLEASE**   | **50.82**   | **63.68**   | **81.75**    | **85.64**    | **87.36**    | **87.76**    |
>
> ---
>
> **W2: Adaptivity and Sensitivity of the Cluster Count ($M$)**
> We clarify that $M$ is not manually fixed at first in FedLEASE. Instead, we only set an upper bound $M_{\text{max}}$ to limit computational and communication costs. The actual number of clusters $M$ is determined automatically during the initialization phase through hierarchical clustering with the silhouette score as the evaluation metric (see §4.1). This procedure allows $M$ to adapt to the underlying client distribution and similarity structure without manual tuning.
>
> Our sensitivity analysis in **Table 8 (Appendix C.2)** demonstrates robustness across different $M_{\text{max}}$ values: when $M_{\text{max}}=2,3,4,8$, our method achieves 85.49%, 87.21%, 87.85%, and 87.76% average performance respectively, with the silhouette-based selection consistently identifying the optimal $M=4$ for our 16-client, 4-task setting.
>
> These results highlight two important observations:
> - The performance degradation when $M_{\text{max}} < M$ (85.49% vs 87.76%) confirms the importance of adequate expert budget. **Notably, even under limited expert capacity, our method still outperforms all baselines**, showcasing its effectiveness in constrained scenarios.
> - Comparable performance between $M_{\text{max}} = 4$ and $M_{\text{max}} = 8$ validates our method’s ability to **avoid over-clustering**, even when given excess capacity.
>
> ---
> We are grateful for the reviewer's constructive suggestions and hope our explanations resolve the raised issues.

---

> > ### Comment · Reviewer_j5zx · 2025-08-05
> >
> > Thank you for your thoughtful responses, which answered my questions well.

---

### Official Review · Reviewer_zQ88 · 2025-07-02

**Clarity:** 2
**Significance:** 3
**Originality:** 3
**Rating:** 4
**Confidence:** 4

**Summary:**

This paper presents FedLEASE, an innovative federated learning framework designed to address the challenges of LoRA-based fine-tuning in heterogeneous client environments. FedLEASE mainly includes two key innovations: optimizing expert allocation by grouping clients based on parameter similarity; and a dynamic top-M mechanism that adaptively determines the optimal number of experts per client, overcoming the limitations of fixed selection strategies.  Extensive experiments demonstrate FedLEASE's superior performance across both natural language understanding (NLU) and generation (NLG) tasks while maintaining communication efficiency.

**Questions:**

See Weaknesses

**Ethical Concerns:**

["NO or VERY MINOR ethics concerns only"]

**Final Justification:**

The authors have added useful results.

**Limitations:**

Yes

**Quality:**

3

**Strengths And Weaknesses:**

**Strengths**
- The paper is well-structured, with clear explanations of the methodology and experiments.
- The proposed adaptive top-M MoE mechanism is innovative and well-justified by empirical results.
- The paper provides the theoretical analysis in the Appendix, ensuring the robustness of the proposed method.
- The ablation studies validate the proposed method's effectiveness and scalability.

**Weaknesses**

- Some parts of the writing in the paper are confusing.
    - First, regarding the adaptive top-M MOE scheme, according to the paper's description, M clusters are obtained through hierarchical clustering, and the protocol of each cluster is used as the initialized LoRA experts. On each client, the LoRA belonging to itself is replicated M times to obtain 2M-1 output routers, thereby allowing each client to dynamically determine the final number of experts used based on its local data. So, what does the "fixed Top-K" mentioned in the paper refer to? Based on the above explanation and my understanding, it should result in K clusters and K LoRA experts, with each client obtaining K output routers. However, in the top-right corner of Figure 5, there are 3 experts and 2 output routers, which is perplexing. Perhaps I missed something, or maybe the confusion arises from inconsistent notation in the method description.

    - Second, regarding the 2M-1 output routers, why is the expert belonging to the client replicated M times? The authors could provide a more intuitive explanation rather than just presenting empirical results. Additionally, each client essentially still selects M experts; the unequal number of experts utilized is due to the replication of its own expert M times. Therefore, I believe attributing the dynamic selection of expert numbers as the reason for achieving optimal personalized performance is not entirely appropriate. The authors might need to adopt a more suitable explanation.

- Concerning the selection of baselines in the experiments. The proposed method has two contributions: one is hierarchical clustering with the silhouette score as the metric to determine the optimal number of experts, and the other is the adaptive top-M MOE. However, the baselines are either FL+LoRA or clustered FL (IFCA)+LoRA, neither of which includes MOE LoRA techniques. Thus, the experimental comparisons are not entirely fair. For example, in the comparison with IFCA+LoRA, it is unclear whether the performance improvement comes from hierarchical clustering, MOE, or the adaptive top-M MOE, making it difficult to quantify these incremental contributions. Nevertheless, it is commendable that the authors demonstrate the advantages of hierarchical clustering for obtaining LoRA experts and the adaptive top-M MOE in Table 2.

- The experimental section only reports the number of parameters, but does not present runtime or communication efficiency. Since the proposed method uses hierarchical clustering to determine the optimal number of experts, this process could be highly time-consuming when the number of clients is large.

- The dataset partitioning in the experiments reveals a natural clustering structure, which explains why the proposed (clustering-based) method achieves strong performance. However, data heterogeneity in FL is typically complex and may exhibit very ambiguous clustering structures. If possible, I would like to know how the proposed method performs under more commonly used Dirichlet partitioning (or other heterogeneous partitioning schemes).

---

> ### Author Rebuttal · Authors · 2025-07-31
>
> Thank you for your thoughtful comments!
>
> **W1: Clarification on "fixed top-$k$ MoE" vs. our proposed "adaptive top-$M$ MoE".**
> To clarify, the term "fixed top-$k$ MoE" in our paper does **not** refer to the number of clusters or LoRA experts being $k$. Instead, it follows the conventional MoE paradigm where, given $M$ experts, each input is routed to the **top-$k$ experts** based on routing scores (i.e., the same fixed number $k$ is used for all data across all clients.)
>
> In contrast, our proposed ''adaptive top-$M$ MoE'' introduces a more flexible and input-adaptive routing mechanism. Each client is first assigned to a primary expert via clustering. Then, the client’s router network $G_i \in \mathbb{R}^{(2M-1) \times d}$ produces **$2M - 1$** routing scores:
> - The **first $M$ scores** correspond to **distinct internal components of the assigned expert**, i.e., they are all routed to the same expert but model different aspects (not different experts).
> - The **remaining $M - 1$ scores** are associated with the other experts.
>
> These $2M - 1$ values are learned via a router network $G_i \in \mathbb{R}^{(2M-1) \times d}$, and selection is conducted via softmax followed by a top-$M$ selection.
>
> To illustrate with an example where $M = 3$ with experts $[E1, E2, E3]$ and $E1$ is the assigned expert, the router may output scores like:
> $[ω₁^{E1}, ω₂^{E1}, ω₃^{E1}, ω₁^{E2}, ω₁^{E3}]$
>
> Here:
> - $ω₁^{E1}$, $ω₂^{E1}$, and $ω₃^{E1}$ are three independently learned scores for the **assigned expert E1**, capturing different aspects of its contribution.
> - $ω₁^{E2}$ and $ω₁^{E3}$ correspond to the other experts.
>
> Depending on the input samples, the output scores can change and the top-M selection might yield:
> - **Sample 1**: ${ω₁^{E1}, ω₂^{E1}, ω₁^{E2}}$ are the top-$M$ highest scores → 2 **unique** experts $E1, E2$ are selected
> - **Sample 2**: ${ω₁^{E1}, ω₂^{E1}, ω₃^{E1}}$ are the top-$M$ highest scores → only 1 **unique** expert $E1$ is selected
> - **Sample 3**: ${ω₂^{E1}, ω₁^{E2}, ω₁^{E3}}$ are the top-$M$ highest scores → all 3 experts are selected
>
> This design has two main benefits:
> 1. It **guarantees participation of the assigned expert** for every client.
> 2. It allows **fine-grained and input-dependent expert allocation**, supporting different levels of specialization across inputs and across layers.
>
> We appreciate the reviewer’s comment and acknowledge that the explanation in the current version could be confusing. In the final version, we will revise the description of this mechanism to make it explicitly clear.
>
> ---
>
> **W2: Choice of MoE + FL baselines.**
> We appreciate the reviewer’s concern regarding baseline selection. To our knowledge, there is currently no prior work that integrates MoE with federated LoRA fine-tuning. Given this gap, we selected recent state-of-the-art FL + LoRA methods for comparison, including:
> - FedIT (ICASSP 2024), which uses single global LoRA
> - FFA-LoRA (ICLR 2024), which freezes LoRA A to reduce server aggregation noise
> - FedSA (ICLR 2025), which applies selective aggregation
> - FedDPA (NeurIPS 2024), which introduces dual-personalized adapters
>
> These methods are all published in top-tier conferences. Besides of these FL + LoRA methods, we also modified a tradtional clustered FL method (i.e. IFCA) with LoRA as an additional baseline. Our goal is to go beyond these works by addressing both (1) **expert allocation** (via clustering), and (2) **expert utilization** (via adaptive top-$M$ routing).
>
> To further isolate and demonstrate the benefit of our adaptive top-$M$ MoE, we conducted direct comparisons in the **“Ablation on Adaptive top-M Mechanism”** section (see **Figure 6(b)**), where we compare our full method against a variant that uses **fixed top-k MoE + adaptive expert allocation**. This ablation serves as the closest proxy to a potential MoE + FL baseline, and results show that our adaptive routing mechanism consistently outperforms all fixed-k alternatives, regardless of the value of $k$.
>
> Additionally, **Table 2** in the main paper provides a clear breakdown of performance contributions from clustering alone (i.e., expert allocation).
>
> ---
>
> **W3: Runtime and clustering computational overhead.**
> Regarding the training and clustering overhead, the clustering step is performed **only once** during the initialization phase and is not repeated during iterative training. Thus, its runtime impact is negligible. As described in §3.3, using only the **LoRA B matrices** provides an efficient and lightweight proxy for task similarity, due to their significantly smaller size compared to full model weights or LoRA A–B combinations.
>
> To further support this, we measured the clustering time as **3.11** seconds on the Intel Xeon Platinum 8570 CPU, which is substantially shorter than the total training time (**193.49** seconds with local training on an NVIDIA B200 GPU). The detailed timing comparisons are provided below:
>
> | Time (s)                        | FedIT | FFA-LoRA | FedDPA | FedSA  | IFCA+LoRA | FedLEASE |
> |---------------------------------|-------|----------|--------|--------|------------|----------|
> | Local Per-Epoch Training Time   | 3.75  | 3.41     | 3.82   | 3.72   | 3.85       | 3.78     |
> | Global Aggregation Time         | 0.048 | 0.038    | 0.051  | 0.042  | 0.061      | 0.055    |
> | Clustering Time | –     | –        | –      | –      | 2.77          | 3.11     |
> | Total Training Time             | 188.70| 171.45   | 192.28 | 187.05 | 263.28     | 193.49   |
>
> We will include the table in the final version to clearly demonstrate that FedLEASE maintains high performance while incurring only modest runtime overhead.
>
> ---
>
> **W4: Dataset partitioning: Task heterogeneity vs. label non-IID.**
> While we acknowledge that traditional federated learning often uses Dirichlet partitioning to simulate label non-IID settings, our task-heterogeneous experimental setup is designed to more accurately reflect real-world scenarios for large language model fine-tuning and training. In practical deployments, clients often hold data from fundamentally different **domains and tasks**, such as:
> - Customer service platforms with dialogue transcripts
> - Educational institutions with exam question-answering datasets
> - Medical research labs with clinical trial summaries
> - Legal departments with contract summarization tasks
> - E-commerce companies with product review classification
>
> Such **task-level heterogeneity** presents a more **challenging and realistic** federated fine-tuning setting (similar setting also considered by FedDPA) than simple label skew within the same task.
>
> We also jointly evaluated our method under **label non-IID conditions**, as shown in **Figure 9** and **Table 7**, where FedLEASE consistently outperforms all baselines even in these skewed scenarios. For these binary classification tasks (e.g., SST-2, MRPC), we followed the **non-IID setup used in prior work** (FedSA, FFA-LoRA) instead of applying Dirichlet partitioning, which may be less meaningful for such binary classification tasks.
>
> ---
>
> We thank the reviewer for these thoughtful comments and hope our detailed clarification addresses each point clearly. We will include these clarifications and results in the final version of the paper.

---

> > ### Comment · Reviewer_zQ88 · 2025-08-03
> >
> > Thank you for the rebuttal. The authors have added useful results on communication/computation costs. While I've reviewed their adaptive top-k MoE approach and baselines, the methodology could be clearer in revision. In addition, including the communication/computation efficiency results in Table 2 would strengthen the paper. Regarding data heterogeneity, additional results beyond task-level heterogeneity would be valuable. For instance, FedSA also employs a Dirichlet(0.5) data partitioning scheme, and other reviewers have also raised concerns about heterogeneity. Currently, I will maintain my score.

---

> > > ### Author Response · Authors · 2025-08-03
> > >
> > > We sincerely thank the reviewer for the constructive feedback. We are pleased that the reviewer found our rebuttal helpful, particularly the additional results on computational overhead, as well as the clarification between fixed top-$k$ MoE and our proposed adaptive top-$M$ mechanism. We will incorporate these results and clarifications into the final version of the paper to further improve its clarity and completeness.
> > >
> > > Regarding the comment on data heterogeneity, we respectfully disagree that our current setup lacks relevance. Our use of task-level heterogeneity, where clients possess data from entirely different tasks (e.g., SST-2, MRPC, QNLI, QQP in GLUE)—better reflects the practical realities of federated fine-tuning in real-world deployments of large pre-trained models. Unlike artificially partitioned datasets with label skew, real clients often work on different downstream applications, making task heterogeneity a **more realistic and challenging** scenario. We have also extended this setting by considering joint heterogeneity, where clients differ not only in task but also in label distribution. As shown in **Figure 9** and **Table 7**, we evaluate FedLEASE under both task-level and label-level heterogeneity. These results demonstrate that our method remains consistently effective and outperforms existing baselines.
> > >
> > > However, we fully appreciate the reviewer’s concern and thus performed additional experiments under **label-only heterogeneity**. To that end, we followed the exact setup used in FedSA under a **Dirichlet($\alpha$=0.5)** partitioning scheme using **20 clients** on the QQP dataset. The following table shows the resulting client label distributions:
> > >
> > > **Table: Client Label Distribution under Dirichlet(0.5) on QQP**
> > >
> > > | Client ID | Class 0 (%) | Class 1 (%) |
> > > |-----------|-------------|-------------|
> > > | 0 | 0.7 | 99.3 |
> > > | 1 | 18.1 | 81.9 |
> > > | 2 | 70.3 | 29.7 |
> > > | 3 | 8.0 | 92.0 |
> > > | 4 | 0.4 | 99.6 |
> > > | 5 | 85.8 | 14.2 |
> > > | 6 | 92.3 | 7.7 |
> > > | 7 | 18.7 | 81.3 |
> > > | 8 | 90.9 | 9.1 |
> > > | 9 | 33.2 | 66.8 |
> > > | 10 | 99.9 | 0.1 |
> > > | 11 | 20.6 | 79.4 |
> > > | 12 | 81.8 | 18.2 |
> > > | 13 | 85.9 | 14.1 |
> > > | 14 | 24.6 | 75.4 |
> > > | 15 | 9.8 | 90.2 |
> > > | 16 | 80.5 | 19.5 |
> > > | 17 | 92.1 | 7.9 |
> > > | 18 | 93.1 | 6.9 |
> > > | 19 | 75.2 | 24.8 |
> > >
> > > **Table: Silhouette Scores**
> > >
> > > | Number of Clusters                        | 2 (best)     | 3      | 4  | 5      | 6      | 7      | 8      |
> > > |------------------------------|--------|--------|----------|--------|--------|--------|--------|
> > > |  Silhouette Score  | **0.1829** | 0.1139 | 0.0553   | 0.0532 | 0.0482 | 0.0527 | 0.0506 |
> > >
> > > **Table: Cluster Groups**
> > >
> > > | Group | Clients |
> > > |-------|---------|
> > > | Group 0 | 2, 5, 6, 8, 10, 12, 13, 16, 17, 18, 19 |
> > > | Group 1 | 0, 1, 3, 4, 7, 9, 11, 14, 15 |
> > >
> > > We then compared the performance of our proposed **FedLEASE** method against the baselines. The results are summarized below:
> > >
> > > **Table: Performance on QQP (Dirichlet-0.5, Label Heterogeneity, 20 Clients)**
> > >
> > > | Method     | Accuracy (%) |
> > > |------------|--------------|
> > > | FedIT      | 83.52          |
> > > | FFA-LoRA   |  83.05         |
> > > | FedDPA     | 85.78          |
> > > | FedSA      | 86.85          |
> > > | IFCA+LoRA  | 84.73          |
> > > | **FedLEASE** | **89.23**    |
> > >
> > > These results demonstrate that **FedLEASE consistently outperforms existing methods** even under label-heterogeneous conditions, further confirming the robustness and generality of our proposed approach.
> > >
> > > Thank you for your reply, and we hope our response addresses your concerns!

---

> > > > ### Comment · Reviewer_zQ88 · 2025-08-05
> > > >
> > > > Thank you for the rebuttal. I will raise my score accordingly. Hope you can include these results in the revised version.

---

### Official Review · Reviewer_Wn44 · 2025-07-02

**Clarity:** 4
**Significance:** 3
**Originality:** 3
**Rating:** 5
**Confidence:** 4

**Summary:**

This paper proposes FedLEASE, a federated LoRA fine-tuning framework that tackles two key challenges in heterogeneous federated learning. FedLEASE clusters clients using the similarity of their LoRA B matrices to determine expert allocation. Additionally, it introduces an adaptive top-M mechanism that enables each client to dynamically select the number of experts to use. Experimental results across various NLU and NLG datasets show that FedLEASE consistently outperforms existing baselines.

**Questions:**

1. How does FedLEASE compare to baseline methods in terms of computational cost and training time?
2. Is the router network maintained locally by each client or distributed from the server? Algorithm 1 indicates that clients reinitialize the router at each communication round.
3. In Table 5, when the number of clients changes (e.g., from 8 to 32), does the optimal number of experts adjust accordingly?
4. In addition to the number of experts each client uses, it would be helpful to show which experts are selected by each client. A discussion on the frequency and distribution of expert selection across clients would provide deeper insights.

**Ethical Concerns:**

["NO or VERY MINOR ethics concerns only"]

**Final Justification:**

The authors have responded thoughtfully to my concerns. In particular, their clarification on computational overhead and training time, which is critical in the context of federated learning, was appreciated. Based on these improvements, I have decided to maintain my score.

**Limitations:**

yes

**Quality:**

4

**Strengths And Weaknesses:**

Strengths
1. The paper identifies two major challenges in federated LoRA fine-tuning and supports them with insightful preliminary experiments.
2. The adaptive top-M routing mechanism enables client-specific expert selection, which improves personalization over fixed top-k strategies.
3. FedLEASE is evaluated on both NLU and NLG tasks, and the paper includes comprehensive experiments that validate the effectiveness and individual contributions of each component.

Weaknesses
1. The clustering phase performs clustering based on the LoRA B matrices from all clients, which could introduce computational overhead, especially in large-scale settings.
2. The ablation studies are not sufficiently thorough. For example, alternative clustering methods are not explored, making it unclear whether Agglomerative Hierarchical Clustering is the optimal choice.

---

> ### Author Rebuttal · Authors · 2025-07-31
>
> Thank you for your valuable comments!
>
> **W1: LoRA B clustering computational overhead.**
> The clustering step is performed **only once** during the initialization phase and is not repeated during iterative training. Thus, its runtime impact is negligible. As observed in §3.3, using only the **LoRA B matrices** offers an efficient and lightweight proxy for task similarity, given their small size compared to full model weights or BA products.
> To further support this, we measured the clustering time (**3.11** seconds on Intel Xeon Platinum 8570 CPU), which is significantly shorter than the total training time (**193.49** seconds with local training on the NVIDIA B200 GPU):
>
> | Time (s)            | FedIT | FFA-LoRA | FedDPA  | FedSA   | IFCA+LoRA | FedLEASE |
> |---------------------|---------------|----------|---------|---------|-----------|----------|
> | Local Per-Epoch Training Time | 3.75          | 3.41     | 3.82    | 3.72    | 3.85      | 3.78 |
> | Global Aggregation Time     | 0.048         | 0.038    | 0.051   | 0.042   | 0.061     | 0.055 |
> | Clustering Time        | -             | -        | -       | -       | 2.77  | 3.11 |
> | Total Training Time     | 188.70    | 171.45 | 192.28 | 187.05 | 263.28 | 193.49 |
>
>
> ---
>
>
> **W2: Alternative clustering methods are not explored.**
> While clustering is indeed an essential component of our proposed method—playing a key role in expert allocation based on client similarity—the specific choice of clustering algorithm is not the focus of our contribution. Our observations (§3.3) suggest that as long as the method captures pairwise similarity between clients (e.g., via LoRA B matrices), the overall performance is relatively robust to the particular clustering strategy.
>
> We adopt Agglomerative Hierarchical Clustering due to its ability to operate directly on pairwise distances without requiring pre-defined centroids. To validate the generality of our approach, we also applied Spectral Clustering, which similarly supports pairwise similarity inputs, and observed comparable performance (within 0.14% accuracy difference).
>
> The tables below demonstrate that both clustering methods achieve similar silhouette scores and downstream performance, reinforcing that our performance gains stem primarily from the expert allocation and adaptive top-$M$ selection mechanisms, rather than the specific clustering algorithm used.
>
> **Silhouette Scores:**
>
> | Method / Number of Clusters                        | 2      | 3      | 4 (best) | 5      | 6      | 7      | 8      |
> |------------------------------|--------|--------|----------|--------|--------|--------|--------|
> | Spectral Clustering           | 0.0585 | 0.0820 | 0.1023   | 0.0739 | 0.0637 | 0.0549 | 0.0218 |
> | Agglomerative Hierarchical   | 0.0599 | 0.0884 | 0.1066   | 0.0965 | 0.0854 | 0.0747 | 0.0660 |
>
> **Final Accuracy Comparison:**
>
> | Method / Task                      | SST-2 | QNLI  | MRPC  | QQP   | Average |
> |----------------------------|-------|-------|-------|-------|---------|
> | Spectral Clustering         | 93.97 | 86.63 | 86.48 | 83.40 | 87.62   |
> | Agglomerative Hierarchical | 93.33 | 87.22 | 86.93 | 83.57 | 87.76   |
>
> ---
>
> **W3: Comparison of computational cost and training time.**
> As discussed in our response to W1, FedLEASE introduces only minimal additional overhead. The clustering step is performed only once during initialization, taking only 3.11 seconds on CPU, which is negligible compared to the total training time (193.49 seconds with local training on GPU, as shown in the above Table in W1).
>
> During training, each client updates only its assigned LoRA expert and router, keeping the per-round computation costs on par with baselines such as FedIT and FedSA.  Importantly, in both GLUE and Flan experiments (**Table 1** and **Table 3** in the main paper), we ensured that the total number of trainable parameters in FedLEASE is comparable to or even smaller than those of baseline methods. Despite this, our method consistently achieves superior performance, validating the efficiency of our expert allocation and adaptive top-$M$ design.
>
> Further, ablation studies (**Table 2** and **Figure 6**) confirm that the adaptive top-$M$ mechanism provides substantial performance gains without increasing training workload, demonstrating both the computational efficiency and effectiveness of our proposed approach.
>
> ---
>
> **W4: Clarification on router reinitialization.**
> Thank you for pointing this out. The router is indeed maintained locally by each client throughout training. The statement in Algorithm 1 suggesting per-round reinitialization was a typo, in fact clients initialize their router only once at the beginning.
>
> We will correct this in the final version of the paper. Additionally, our ablation study in **Figure 6(a)** explicitly demonstrates that maintaining the router locally leads to better performance, supporting our design choice.
>
> ---
>
> **W5: Does the optimal number of experts vary with number of clients?**
> In the experiments where the number of clients varies, the underlying data distribution remains fixed across four distinct datasets (SST-2, QNLI, MRPC, QQP). As a result, our clustering procedure consistently identifies 4 clusters, which aligns with the task/domain-level heterogeneity rather than the client count.
>
> Importantly, the clustering is based on pairwise similarity of LoRA B matrices, a core component of our method motivated by prior observations that LoRA B captures task-specific information (§3.3). Overall, the number of experts is determined by the **intrinsic task-level differences** across clients. If additional tasks or datasets are introduced, our clustering procedure would naturally detect a different number of clusters as needed.
>
> ---
>
> **W6: Frequency and distribution of expert selection.**
> We provide additional analysis showing **normalized expert selection percentage** at different layers for each client. This allows us to capture relative expert usage, especially when clients assign multiple experts.
>
> Our findings reveal several important patterns that aligns with Figure 6(c) in our paper:
>
> - **Deeper layers** (e.g., Layer 23) show strong reliance on the expert corresponding to the client’s assigned cluster, indicating more specialized and task-specific behavior as depth increases.
> - **Shallower layers** (e.g., Layer 0) tend to involve a more diverse set of experts, reflecting greater need for knowledge transfer across domains in earlier representations.
> - Despite local updates of the router networks clients within the same cluster show highly similar expert selection patterns, validating the effectiveness of the initial clustering based on LoRA B similarity.
>
> Thus we can find that the number of experts selected varies across layers—shallow layers often draw from more experts than deeper ones. This confirms the flexibility and effectiveness of our adaptive expert routing mechanism. Unlike traditional fixed top-$k$ MoE strategies—which constrain every data sample, every layer to a fixed number of experts—our method enables fine-grained adaptivity across samples and layers, offering a much more flexable approach to expert utilization in federated fine-tuning.
>
> | Client ID | Cluster Group | Layer 0                             | Layer 11                            | Layer 23                            |
> |-----------|---------------|-------------------------------------|-------------------------------------|-------------------------------------|
> | 0         | 2          | [0.146, 0.157, 0.602, 0.095]        | [0.031, 0.011, 0.932, 0.026]        | [0.001, 0.003, 0.993, 0.002]        |
> | 1         | 2          | [0.134, 0.143, 0.632, 0.090]        | [0.033, 0.012, 0.927, 0.028]        | [0.001, 0.003, 0.994, 0.002]        |
> | 2         | 2         | [0.129, 0.139, 0.642, 0.089]        | [0.033, 0.012, 0.925, 0.029]        | [0.001, 0.003, 0.993, 0.002]        |
> | 3         | 2  | [0.131, 0.141, 0.638, 0.090]        | [0.031, 0.014, 0.927, 0.028]        | [0.001, 0.003, 0.994, 0.002]        |
> | 4         | 0  | [0.514, 0.158, 0.166, 0.162]        | [0.875, 0.082, 0.015, 0.029]        | [0.885, 0.073, 0.014, 0.028]        |
> | 5         | 0    | [0.513, 0.158, 0.166, 0.162]        | [0.877, 0.082, 0.013, 0.029]        | [0.884, 0.075, 0.014, 0.027]        |
> | 6         | 0    | [0.505, 0.160, 0.169, 0.165]        | [0.872, 0.085, 0.014, 0.029]        | [0.884, 0.072, 0.015, 0.029]        |
> | 7         | 0       | [0.510, 0.159, 0.168, 0.164]        | [0.876, 0.082, 0.014, 0.028]        | [0.885, 0.075, 0.013, 0.028]        |
> | 8         | 3             | [0.133, 0.102, 0.083, 0.682]        | [0.057, 0.079, 0.083, 0.780]        | [0.072, 0.031, 0.134, 0.763]        |
> | 9         | 3     | [0.124, 0.100, 0.079, 0.696]        | [0.057, 0.077, 0.084, 0.783]        | [0.073, 0.031, 0.136, 0.760]        |
> | 10        | 3             | [0.133, 0.102, 0.083, 0.682]        | [0.056, 0.079, 0.083, 0.782]        | [0.074, 0.031, 0.134, 0.762]        |
> | 11        | 3             | [0.131, 0.101, 0.082, 0.685]        | [0.056, 0.078, 0.083, 0.783]        | [0.074, 0.031, 0.133, 0.763]        |
> | 12        | 1             | [0.098, 0.618, 0.181, 0.103]        | [0.150, 0.795, 0.043, 0.011]        | [0.027, 0.878, 0.006, 0.089]        |
> | 13        | 1             | [0.098, 0.617, 0.182, 0.103]        | [0.148, 0.798, 0.041, 0.012]        | [0.028, 0.878, 0.005, 0.090]        |
> | 14        | 1             | [0.097, 0.631, 0.172, 0.100]        | [0.148, 0.794, 0.045, 0.013]        | [0.028, 0.877, 0.006, 0.089]        |
> | 15        | 1             | [0.099, 0.614, 0.184, 0.104]        | [0.151, 0.799, 0.038, 0.011]        | [0.026, 0.876, 0.005, 0.093]        |
> ---
>
> We thank the reviewer for the valuable comments and will incorporate these clarifications and additions in the final version.

---

> > ### Comment · Reviewer_Wn44 · 2025-08-04
> > **Thanks for the response**
> >
> > Thank you for the detailed response, the author effectively addressed my issue, I will maintain my grade. Hope you can add these details to your appendix.

---

### Comment · Area_Chair_aaw1 · 2025-08-06

Dear reviewers and authors,

The discussion period is ending soon, so please make sure to finalize the discussion (and make mandatory acknowledgments, if not completed yet). Thank you so much for your active participation!

Best,
AC.

---

### Note · Authors · 2025-08-13

We sincerely thank all reviewers for their recognition of our work and for the valuable, constructive feedback provided throughout the review process. We are pleased to see that, after the rebuttal and discussions, all reviewers expressed positive evaluations and no remaining concerns, which we believe reflects a shared understanding of the contributions and significance of our paper.

Our work addresses two fundamental questions in federated fine-tuning with LoRA:
1. Given heterogeneous client distributions, what is the optimal number of LoRA experts to allocate, and how should clients contribute to their training?
2. Given allocated experts, how can each client dynamically determine the optimal number of experts to use based on its data characteristics?

To address these challenges, we propose FedLEASE, a framework that initializes LoRA experts in a data-driven manner and incorporates an adaptive top-$M$ mechanism, enabling efficient and personalized expert utilization across clients.

In the final version, we will incorporate all the promised improvements and clarifications made during the rebuttal phase, including:
* Placing detailed implementation information in main paper instead of appendix and standardizing all citations to official published versions (e.g., citing the NeurIPS 2024 version of FedDPA) to enhance readability and citation quality.
* Providing additional runtime and communication efficiency comparisons to comprehensively assess system overheads.
* Clarifying the adaptive top-$M$ mechanism with an explicit numerical example, and emphasizing that the number of clusters $M$ is determined automatically with only an upper bound $M_{\text{max}}$ set for efficiency control.
* Including a table and discussion on expert selection frequency and distribution across clients.
* Adding a label non-IID only experiment with Dirichlet $\alpha = 0.5$ to further validate robustness under heterogeneous label distributions.

We are grateful for the reviewers’ careful reading, constructive suggestions, and the time invested in evaluating our work. We believe these refinements will further enhance the clarity, completeness, and impact of the paper.

---

### Decision · Program_Chairs · 2025-09-17

**Decision:**

Accept (poster)

**Comment:**

This paper introduces FedLEASE, a federated LoRA fine-tuning algorithm powered by mixture of adaptive number of LoRA experts. To construct multiple experts, FedLEASE clusters the LoRA from clients. The reviewers found the proposed method to be well-motivated and effective, providing a meaningful advancement over the prior art. The main criticisms have been on relatively weak efficiency analysis, ablation studies, plus the experimental setup. During the rebuttal period, the authors have addressed the concerns comprehensively, and now reviewers are unanimously positive toward the acceptance of the manuscript.